# Diagnostic evaluation of river discharge into the Arctic Ocean and its impact on oceanic volume transports

Susanna Winkelbauer[1], Michael Mayer[1,2], Vanessa Seitner[1], Ervin Zsoter[2], Hao Zuo[2], and Leopold Haimberger[1]

[1]Department of Meteorology and Geophysics, University of Vienna, Vienna Austria
[2]European Centre for Medium-Range Weather Forecasts, Reading, United Kingdom

**Correspondence:** Susanna Winkelbauer (susanna.winkelbauer@univie.ac.at)

**Abstract.** This study analyses river discharge into the Arctic Ocean using state-of-the-art reanalyses such as the fifth-generation European Reanalysis (ERA5) and the reanalysis component from the Global Flood Awareness System (GloFAS). GloFAS, in its operational version 2.1, combines the land surface model (Hydrology Tiled ECMWF Scheme for Surface Exchanges over Land, HTESSEL) from ECMWF's ERA5 with a hydrological and channel routing model (LISFLOOD). Further, we analyse

GloFAS' most recent version 3.1, which is not coupled to HTESSEL but uses the full configuration of LISFLOOD.

Seasonal cycles, as well as annual runoff trends are analysed for the major Arctic watersheds - Yenisei, Ob, Lena and Macken-zie - where reanalysis-based runoff can be compared to available observed river discharge records. Further, we calculate river discharge over the whole Pan-Arctic region and, by combination with atmospheric inputs, storage changes from the Gravity Recovery and Climate Experiment (GRACE) and oceanic volume transports from ocean reanalyses and assess closure of the

non-steric water volume budget. Finally, we provide best estimates for every budget equation term using a variational adjust-ment scheme.

Runoff from ERA5 and GloFAS v2.1 feature pronounced declining trends, induced by two temporal inhomogeneities in ERA5's data assimilation system, and seasonal river discharge peaks are underestimated by up to 50% compared to obser-vations. The new GloFAS v3.1 product exhibits distinct improvements and performs best in terms of seasonality and long term

means, however opposing to gauge observations it also features declining runoff trends. Calculating runoff indirectly through the divergence of moisture flux is the only reanalysis-based estimate that is able to reproduce the river discharge increases mea-sured by gauge observations (Pan-Arctic increase of 2% per decade). In addition we examine Greenlandic discharge, which contributes about 10% of the total Pan-Arctic discharge and features strong increases mainly due to glacial melting.

The variational adjustment yields reliable estimates of the volume budget terms on an annual scale, requiring only moderate

adjustments of less than 3% for each individual term. Approximately $6583\pm84$ km$^3$ freshwater leave the Arctic Ocean per year through its boundaries. About two thirds of this are contributed by runoff from the surrounding land areas to the Arctic Ocean ($4379\pm25$ km$^3$ per year) and about one third is supplied by the atmosphere. However, on a seasonal scale budget residuals of some calendar months were too large to be eliminated within the a priori spreads of the individual terms. This suggests that systematical errors are present in the reanalyses and ocean reanalyses data-sets, which are not considered in our uncertainty

estimation.

## 1   Introduction

Rapid surface warming in the Arctic region has strong impacts on the Arctic water balance and its individual hydrological components, almost certainly leading to an amplification in runoff, evapotranspiration and precipitation (Rawlins et al., 2010;
Collins et al., 2014). Increasing river discharge and precipitation trends and intensified sea ice melt coupled with an increase of freshwater inflow through Bering Strait lead to an increase of liquid freshwater stored in the Arctic Ocean (Morison et al., 2012; Haine et al., 2015; Haine, 2020). Ultimately, this could result in enhanced southward exports of low-density waters (Lin et al., 2021) into the Atlantic Ocean, impacting the oceanic circulation also on a global scale. Altogether the hydrological cycle is a complex process with tight coupling between the individual components, having impacts on energy and mass budgets and
eventually sea level rise, both regionally (Proshutinsky et al., 2001; Moon et al., 2018) and globally (e.g., Box et al., 2018). Therefore, the quantification of the individual hydrological components and their changes is of great importance.

With the Arctic Ocean being almost entirely surrounded by landmasses and some of the world's largest rivers draining into it, the link between ocean and surrounding land is strong. Hence, runoff forms one of the key variables in the Arctic freshwater budget. However, direct quantification of river discharge into the Arctic Ocean is aggrevated by the fact that about 30-40% of
the Pan-Arctic drainage area is now unmonitored (Shiklomanov et al., 2002). Hydrological monitoring suffered a widespread decline from 74% in 1986 to 67% by 1999 - in Siberia even 73% of river gauges were closed between 1986 and 1999 (Shiklomanov et al., 2002; Shiklomanov and Vuglinsky, 2008). In addition, significant portions of the rivers' discharge may bypass the gauging stations through braided channels or as submarine groundwater. Further, climatological conditions pose a hindrance to gauge measurements, as temperatures in the northern latitudes often lead to river freeze up in late autumn and flooding in
spring due to river-ice break up (Syed et al., 2007).

Atmospheric reanalyses produce gridded estimates of atmospheric and land components, providing spatially continuous estimates of variables such as runoff. They represent highly useful tools for climate monitoring. In this study we evaluate runoff from state-of-the art reanalyses and GloFAS to provide a best estimate of Pan-Arctic river discharge and to incorporate it into the Arctics freshwater and volume budget. However, data assimilation systems can introduce biases and temporal discontinu-
ities, as changes in the observing system are sometimes inevitable and may lead to inhomogeneities in the time series. One known change is the introduction of the IMS (Interactive Multisensor Snow and Ice Mapping System) snow product in ERA5, which led to a negative shift in ERA5's snowmelt and consequently also runoff (Hersbach et al., 2020; Zsótér et al., 2020).

The first complete freshwater budget for the Arctic Ocean is proposed by Aagaard and Carmack (1989) and updated by Serreze et al. (2006) and Dickson et al. (2007). Since then, the amount of available data, in particular of atmospheric and oceanic re-
analyses, opened new possibilities for evaluation of the coupled oceanic and atmospheric energy and hydrological cycles. For example, Mayer et al. (2019) have presented a substantially improved depiction of the Arctic energy cycle compared to earlier assessments. With regard to freshwater or the water volume budget, the data situation has improved as well. New collections

of hydrological data of the far north have been published (Shiklomanov et al., 2021b). Tsubouchi et al. (2012, 2018) presented observation-based estimates of volume fluxes through Arctic gateways. This opens the opportunity to overspecify the Arctic volume budgets and thus also to give residual and bias estimates.

This paper is structured as follows. The next section describes the used data and presents the study domain, followed by the methodology. The results are presented in Sect. 4 and are subdivided into seasonal cycles and trends for the four major Arctic watersheds (Sect. 4.1), Pan-Arctic seasonalities and trends (Sect. 4.2) and an assessment of budget closure by comparison with oceanic volume fluxes (Sect. 4.3). Section 5 presents conclusions and in the appendix a list of acronyms used throughout the text can be found.

## 2   Data and study domain

Runoff is taken from the European Centre for Medium-Range Weather Forecast's (ECMWF) 5th generation global climate reanalysis ERA5 (Hersbach et al., 2020), as well as from its offline simulation ERA5-Land and is downloaded through the Copernicus Climate Change Service (C3S) Climate Data Store (Hersbach et al., 2019; Muñoz Sabater, 2019). Runoff from ERA5 and ERA5-Land are both produced by the land-surface model Hydrology Tiled ECMWF Scheme for Surface Exchanges over Land (HTESSEL, Balsamo et al. (2009)) of the ECMWF Integrated Forecasting System (IFS). Contrary to ERA5, ERA5-Land is not coupled to the atmospheric model of the IFS and no direct data assimilation is used, however observations have an indirect influence, as atmospheric variables from ERA5 are used as atmospheric forcing in ERA5-Land. An advantage of ERA5-Land is the enhanced global resolution of 9 km (31 km for ERA5) (Muñoz Sabater et al., 2021a). Runoff data are converted into river discharge (liquid water and ice), by integration over the associated catchment area. We also look into the runoff climatology BT06 (Bourdalle-Badie and Treguier, 2006), that is used in the global ocean-ice reanalysis ORAS5.

Furthermore, we consider the Global Flood Awareness System (GloFAS) river discharge reanalysis. GloFAS is developed by ECMWF and the Joint Research Centre (JRC), as part of the Copernicus Emergency Management Service (CEMS). GloFAS is publically available with data accessible from the Copernicus Climate Change Service Climate Data Store (CDS). Its operational version, GloFAS 2.1 (Harrigan et al. (2019), hereafter denoted $GloFAS_{E5}$), combines a simplified version of the hydrological river routing model LISFLOOD (Knijff et al., 2010), to simulate groundwater processes and river routing, with runoff data from HTESSEL, the land surface model used in ERA5 (Harrigan et al., 2020). In addition we examine an experimental GloFAS version, that also uses LISFLOOD's channel routing, but forces it with runoff from ERA5-Land - hereafter denoted $GLOFAS_{E5L}$ - and GloFAS version 3.1 ($GLOFAS_{E5_{new}}$), which uses the full configuration of the LISFLOOD model and is not coupled to HTESSEL but rather produces its own runoff by using directly precipitation, evaporation and temperature from ERA5. While $GLOFAS_{E5}$ and $GLOFAS_{E5_{new}}$ are available from 1979 to near real time, the experimental version $GLOFAS_{E5L}$ is only available from 1999 to 2018.

Data from ERA5 and GloFAS are compared to available observed river discharge records. Observing records vary among the different countries and rivers, with the longest time series coming from Russia, where discharge monitoring began in the mid 1930s. In contrast, discharge measurement in North America did not begin until the 1970s (Holmes et al., 2018). The data used

|           | Gauges                | GloFAS                |
| --------- | --------------------- | --------------------- |
| Yenisei   | 67.48°N; 86.50°E      | 67.45°N; 86.45°E      |
| Ob        | 66.57°N; 66.53°E      | 66.55°N; 66.45°E      |
| Lena      | 70.70°N; 127.65°E     | 72.25°N; 126.75°E     |
| Mackenzie | 67.45°N; -133.75°E    | 67.45°N; -133.75°E    |

**Table 1.** Positions of gauge observations and GloFAS locations for Ob, Yenisei, Lena and Mackenzie.

in this study comes from Roshydromet (Ob, Yenisei and Lena) and from the Water Survey of Canada (Mackenzie) and was downloaded through the Arctic Great Rivers Observatory (Shiklomanov et al., 2021b). Table 1 shows coordinates of gauge observations and GloFAS sampling locations for Yenisei, Ob, Lena and Mackenzie. For the Pan-Arctic approach, river discharge from additional 20 rivers was taken. Gauging records for Kolyma, Severna Dvina, Pechora and Yukon are available for our pe-

riod of interest 1979-2019 and are also downloaded through the Arctic Great Rivers Observatory (Shiklomanov et al., 2021b), while records for 16 further rivers (Pur, Taz, Khatanga, Anabar, Olenek, Yana, Indigirka, Alazeya, Anadyr, Kobuk, Hayes, Tana, Tuloma, Ponoy, Onega, Mezen) are taken from the Regional Arctic Hydrographic Network data set (R-ArcticNET, Lammers et al., 2001) for the period 1979-1999. We calculated an observation-based Pan-Arctic river discharge for the whole period of 1979 to 2019, by calculating discharge separately for every time step (= every month), using all river discharge measurements

available at those time steps. The total Pan-Arctic discharge is then obtained by calculating river discharge for the ungauged area at each individual timestamp (using two different calculation methods - see section 4.2) and adding it to the observed discharge.

Atmospheric components like precipitation, evaporation, atmospheric storage change and the divergence of moisture flux (VIWVD) are taken from ERA5 and in Sect. 4.3 we additionally use VIWVD data from the Japanese 55-year Reanalysis JRA55

(Kobayashi et al., 2015) and JRA55-C (Kobayashi et al., 2014), which only assimilates conventional observation data. Land storage is derived from snow depth (given as water equivalent) and soil water changes from ERA5. Groundwater storage is not represented in ERA5 and ERA5-Land and also the representation of frozen land components is not ideal in HTESSEL, as glaciers are depicted as large amounts of snow which are kept fixed to 10 m of snow water equivalent. When melting conditions are reached, the snow produces a water influx to the soil and consequently contributes to the total runoff. However, the mass

balance is not accounted for over glaciers as the snow is restocked to constantly stay at the fixed 10 m level and hence changes in the glacial storage component cannot be assessed properly. The soil water content includes liquid as well as frozen components and thus also includes permafrost. When the soil temperature reaches melting conditions, the soil water contributes to sub-surface runoff and the soil water storage declines. However, a recent study by Cao et al. (2020) concluded that ERA5-Land soil data are not optimal for permafrost research, due to a warm bias in soil temperature that leads to an overestimation of the

active-layer thickness and an underestimation of the near-surface permafrost area. Therefore, we additionally include satellite data from GRACE (Gravity Recovery and Climate Experiment, Tapley et al. (2004)) and GRACE Follow-On (Landerer et al., 2020), as land water storage from GRACE includes changes in soil moisture (including permafrost), glaciers, snow, surface

| Product | Description | Variable | Period | |
|---|---|---|---|---|
| ERA5 | Fifth generation ECMWF reanalysis using IFS (+ HTESSEL) | Runoff [m/s] | 1979-2019 (back extension to 1950 available) | Hersbach et al. (2020) |
| ERA5-Land | Offline simulation of ERA5 without DA using HTESSEL | Runoff [m/s] | 1981-2019 (back extension to 1950 expected in 2021) | Muñoz Sabater et al. (2021b) |
| GloFAS$_{E5}$ | ERA5 runoff + simplified LISFLOOD | River discharge [m$^3$/s] | 1979-2019 | Harrigan et al. (2020) |
| GloFAS$_{E5L}$ | ERA5-Land runoff + simplified LISFLOOD | River discharge [m$^3$/s] | 1999-2018 | - |
| GloFAS$_{E5new}$ | Full configuration of LISFLOOD | River discharge [m$^3$/s] | 1979-2019 | - |
| BT06 | Runoff climatology used in ORCA025 | River discharge [m$^3$/s] | Climatology | Bourdalle-Badie and Treguier (2006) |
| Observations | Measurements at gauging stations | River discharge [m$^3$/s] | - | - |

**Table 2.** List of all runoff and river discharge sources

| Straits | ORCA |
|---|---|
| Fram | 78.80°-78.80°N,20.60°W-11.50°E |
| Davis | 66.60°-67.30°N, 61.20°-54.00°W |
| Bering | 65.90°-65.70°N, 170.00°-168.30°W |
| BSO | 77.40°-69.70°N, 18.00°-20.40°E |
| Hecla and Fury | 69.85°-70.00°N, 84.50°-84.32°W |

**Table 3.** Start and end points of the sections used for lateral flux calculations on the native ORCA grid.

water, aquifers and groundwater. Also oceanic storage terms are calculated using ocean bottom pressure changes from GRACE. Monthly ocean bottom pressure anomalies and land mass anomalies for the period of April 2002 to December 2019 are derived

from time-variable gravity observations and are given as equivalent water thickness changes. Due to its limited spatial resolution GRACE data is prone to signal leakage from land to ocean, hence we use Mascon solutions (mass concentration blocks, Watkins et al. (2015)) which reduce the leakage effect. We examine three Mascon solutions, RL06 v02 from the Center for Space Research at University of Texas, Austin (CSR, Save (2020); Save et al. (2016)), RL06 from the NASA Jet Propulsion Laboratory (JPL, Wiese et al. (2018); Watkins et al. (2015)) and RL06 from the Goddard Space Flight Center (GSFC, Loomis

et al. (2019)) and estimate ocean and land water storage components by taking the mean of those three solutions.

Oceanic volume fluxes through the main Arctic Gateways are calculated by integrating the cross-sectional velocity component around the Pan-Arctic boundary from the Copernicus Marine Environment Monitoring Service (CMEMS) Global ocean Reanalysis Ensemble Product (GREP, Desportes et al., 2017; Storto et al., 2019), an ensemble of four global ocean reanalyses for the period from 1993 to present. GREP consists of current ocean reanalysis efforts from the Centro Euro-Mediterraneo sui

Cambiamenti Climatici (CGLORS, Storto and Masina, 2016), the UK Met Office (FOAM, Blockley et al., 2014), Mercator Ocean (GLORYS, Garric et al., 2017)) and the ECMWF (ORAS5, Zuo et al., 2018, 2019). While the GREP ensemble members use the same ocean modeling core and atmospheric forcing (ERA-Interim, Dee et al., 2011), there are differences in used observational data and data assimilation techniques, as well as in the reanalysis initial states, NEMO (Nucleus for European Modelling of the Ocean) versions, the sea-ice models, physical and numerical parametrizations and air-sea flux formulations.

The data assimilation methods differ in many points, including the deployed assimilation schemes which range from 3DVAR (three-dimensional variational data assimilation) to SEEK (Singular Evolutive Extended Kalman Filter). Furthermore, there are differences in the input observational dataset, in surface nudging, in the time-windows for assimilation and analysis as well as in the applied bias correction schemes. All those differences lead to an important dispersion between the reanalysis implementations and add up to the ensemble spread (Storto et al., 2019). For further details we refer to Storto et al. (2019)

as well as the individual data documentations. In addition, volumetric fluxes are derived from moorings within the so-called ARCGATE project (Tsubouchi et al., 2019), covering the period from October 2004 to May 2010. These observation-based estimates of volume fluxes come from a mass-consistent framework that views the Arctic Ocean as a closed box surrounded by landmasses and hydrographic observation lines located in the four major Arctic gateways. The hydrographic lines consist of arrays of moored instruments measuring variables temperature, salinity and velocity, making it possible to calculate fluxes

of volume, heat and freshwater. For more details about the framework see Tsubouchi et al. (2012, 2018).

Figure 1 presents the study domain. As there is no strict boundary to the south, the definition of the Arctic's geographic extent varies in past studies and there is no general rule whether to include Greenland and the Hudson Bay or not. We chose our study domain to be consistent with Tsubouchi et al. (2012) as we wanted to compare the oceanic fluxes from ocean reanalysis with the observation-based estimates from the ARCGATE project. The Arctic Ocean is bounded by the position of hydrographic

moorings in the main gateways. Bering Strait forms the only passage to the Pacific Ocean and delivers low salinity waters into the Arctic, while liquid freshwater and sea ice leave the Arctic Ocean mainly through Fram and Davis Strait. The fourth major strait is the Barents Sea Opening (BSO), where high salinity waters from the Atlantic Ocean are imported into the Arctic.

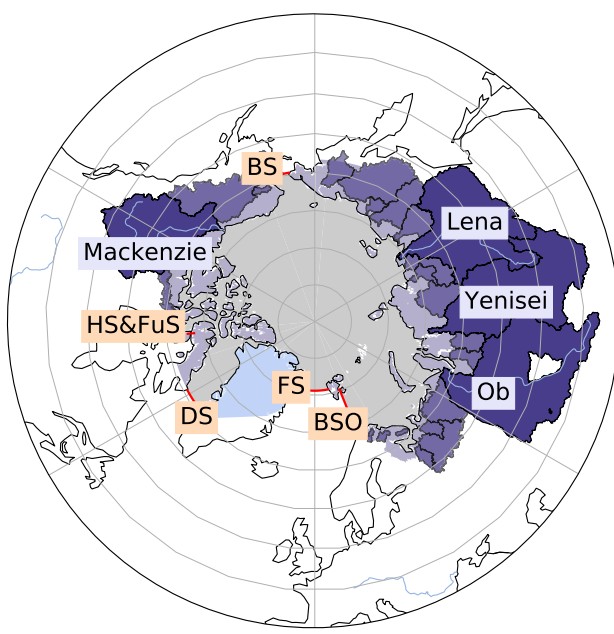

**Figure 1.** Map of the main study area, consisting of the oceanic area bounded by moorings in Bering Strait (BS), Davis Strait (DS), Fram Strait (FS), the Barents Sea Opening (BSO) as well as Hecla and Fury Strait (HS & FuS) (indicated by grey shading; corresponds to $11.3 \times 10^6$ km $^2$) and the land area draining into it (purple shading; corresponds to $18.2 \times 10^6$ km $^2$ for mainlands and islands and additional $0.95 \times 10^6$ km$^2$ for Greenland). The colour grading of the land areas indicates the four largest river catchments (Ob, Yenisei, Lena and Mackenzie; dark-purple), the additional 20 catchments with observations and the smaller catchments and coastal areas where no observations were available (light-purple).

Furthermore, there are two small passages, Fury and Hecla Strait, that connect the Arctic Ocean with Hudson Bay. The terrestrial domain consists of land areas draining into the Arctic Ocean, including the Canadian Arctic Archipelago (CAA)
as well as islands along the Eurasian coast. At the Pacific passage Yukon and Anadyr rivers are considered, as they represent important sources for inflow of low salinity waters into the Arctic Ocean through Bering Strait. The total oceanic and terrestrial areas correspond to $11.3 \times 10^6$ km$^2$ and $18.2 \times 10^6$ km$^2$ respectively. For volume budget analyses we further incorporate Greenlandic discharge and storage change north of Davis and Fram Strait, which adds an additional terrestrial catchment area of $0.95 \times 10^6$ km$^2$. The outlines for the individual river catchments were taken from the CEO Water Mandate Interac-
tive Database of the World's River Basins (http://riverbasins.wateractionhub.org/) and regional outlines like e.g., the Canadian Arctic Archipelago were taken from the Global Runoff Data Centre (GRDC, 2020).

## 3 Methods

### 3.1 Budget equations

A common way to calculate the oceanic freshwater budget is through the assumption of a reference salinity (e.g., Serreze et al.,

2006; Dickson et al., 2007; Haine et al., 2015). However, the outcome is dependent of those reference salinities in a nonlinear way, so that slight differences in the choice of the reference value lead to very different estimates of freshwater transports, both temporally and spatially. Hence, Schauer and Losch (2019) declared freshwater fractions not useful for the analysis of oceanic regions and rather recommend the usage of salt budgets for salinity assessments. In this paper we do not calculate salt budgets, but we estimate volume budgets, and hence also avoid the usage of a reference salinity. Hereinafter, the volumetric budget

equations for atmosphere, land and ocean are formulated.

*Atmosphere*

The change of water storage in the atmosphere - here expressed as water vapor integrated from the earth's surface to the top of the atmosphere (i.e. total column water vapor; hereafter denoted $S_A$) - denotes the left side of the volumetric budget equation. Atmospheric liquid water and ice are neglected, as they represent only a very small fraction of water in comparison

to atmospheric water vapor and lateral moisture fluxes. Generally atmospheric water in liquid and solid phase are only present in significant amounts in regions with high tropical cumulus clouds and over warm ocean currents (Serreze and Barry, 2014). The atmospheric storage change ($S_A$) is balanced by the surface freshwater fluxes evapotranspiration ET and precipitation P (both in SI units ms$^{-1}$) and the vertically integrated horizontal moisture flux divergence (last term; hereafter denoted as VIWVD):

$$\frac{1}{g\rho_w} \frac{\partial}{\partial t} \int_0^{p_s} q \, dp = ET - P - \frac{1}{g\rho_w} \boldsymbol{\nabla} \cdot \int_0^{p_s} q\boldsymbol{v} \, dp \quad (1)$$

with the gravitational constant $g$, surface pressure $p_s$, specific humidity $q$, density of freshwater $\rho_w$ and the horizontal wind vector $\boldsymbol{v}$. The equation above is probably more familiar to the reader as a mass budget equation, without $\rho_w$ in the denominator. In order to get volumetric water fluxes, we divided by the density of freshwater $\rho_w$=1000 kg m$^{-3}$, which is assumed constant in this paper, neglecting dependence on temperature and soluble substances. Further, all terms are integrated over the total Arctic

area $A_{total}$ (including the Arctic Ocean and all terrestrial catchments) to obtain SI units of $m^3 s^{-1}$. For presentation of results, we will often use Sverdrup $Sv$ (=1e6 m$^3$s$^{-1}$), milli-Sverdrup $mSv$ (=1e3 m$^3$s$^{-1}$) or km$^3yr^{-1}$ as more convenient units.

*Land*

The change in land water storage ($S_L$) can be expressed as sum of changes of volumetric soil water $SWVL_n$ integrated over the corresponding soil depth $l_n$, changes in snow depth SD - given as snow water equivalent - and changes in groundwater

storage $GS$. Changes in land water storage are balanced through precipitation $P_L$ and evapotranspiration $ET_L$ over land and runoff R (in SI units $ms^{-1}$). To obtain volumetric fluxes we again perform areal integration over the corresponding area (here

the land area $A_l$).

$$\frac{\partial}{\partial t}\left(SD + \int_{l_n} SWVL_n\,dl_n + GS\right) = P_L - ET_L - R \tag{2}$$

Rearranging Eq. (2) we obtain an estimate of the water discharging into the Arctic Ocean independent of runoff itself. As we

prefer analyzed quantities we can further insert Eq. (1) to substitute $P_L$ and $ET_L$, which are derived from short-term forecasts from ERA5, through the analyzed quantities $VIWVD_L$, and atmospheric storage change:

$$R = -\frac{\partial S_L}{\partial t} + P_L - ET_L = -\frac{\partial S_L}{\partial t} - \frac{\partial S_A}{\partial t} - VIWVD \tag{3}$$

*Ocean*

Following Bacon et al. (2015) the oceanic mass budget equation can be expressed as

$$\iiint_V \frac{\partial \rho}{\partial t}\,dV = F^{surf} - \iint_{A_{sz}} \rho\,\boldsymbol{c}\cdot\boldsymbol{n}\,ds\,dz \tag{4}$$

The left side of Eq. (4) denotes the change in mass over a closed volume, $F^{surf}$ describes any surface mass input/output in the form of precipitation, evaporation and runoff. The last term of Eq. (4) denotes ice and ocean side-boundary fluxes from or into the volume, with the horizontal sea water velocity $\boldsymbol{c}$ and integration along the along-boundary coordinate s and depth z. Further, we apply the Boussinesq approximation and assume $\rho$ as constant. We adopt the reference density used in the Nucleus

for European Modelling of the Ocean (NEMO) ocean model of $\rho_O$=1035 kg/m$^3$ (Madec et al., 2019) and divide Eq. (4) by $\rho_O$. This yields an expression where steric effects are ignored and only volume, also considered as Boussinesq mass ($M_O = \rho_O V$), is conserved (Madec et al., 2019; Bacon et al., 2015):

$$\frac{\partial}{\partial t}\iiint_V dV = P_O - ET_O + R - \iint_{A_{sz}} \boldsymbol{c}\cdot\boldsymbol{n}\,ds\,dz \tag{5}$$

The change of oceanic volume (derived from bottom pressure changes; hereafter denoted $S_O$) is balanced by precipitation

and evapotranspiration over the oceanic domain ($P_O$ and $ET_O$), runoff from the land domain $R$ and further volume can leave and enter the ocean laterally over its vertical boundaries, described by the last term of the equation (oceanic lateral transport, denoted F). The liquid portion of F is calculated by integrating the cross-sectional velocity component along the side areas of the Arctic boundary. Additionally, we add ice transports, which are calculated analogously by integrating the cross-sectional ice velocity over the grid-point-average ice depth and integrating it over the Arctic boundary. As volume exchange between

liquid ocean and sea-ice is conserved in the NEMO model, we additionally remove the liquid water volume that is actually replaced by sea ice, which we call the equivalent liquid water flux. The equivalent liquid water flux at a given grid point is calculated by integrating the liquid volume flux over the grid-point-average ice depth and taking 90% of the result (as only 90% of the icebergs are underwater). As ice velocities from the public CMEMS data portal are only available from two of the ocean reanalyses (ORAS5 and GLORYS2V4) we calculate the ice flux "correction" term for the GREP ensemble by taking

the mean of those two products. However, as the impact of the correction is quite similar for ORAS5 and GLORYS2V4 we

believe that the correction is accurate enough for the purpose of this study. Changes in ocean density do not affect the volume as the steric effect is missing due to the Boussinesq approximation.

In this paper we mostly present monthly means, derived by averaging the corresponding fields from reanalyses in their native temporal resolution. Horizontal interpolation and vertical interpolation has been avoided by using all reanalysis products in their native grid representation. Care has been taken also to average over the same area for all products as far as this is possible. The lateral volume fluxes through the ocean gateways were evaluated along paths on the native ORCA grid that followed the ARCGATE mooring arrays as closely as possible - ORCA coordinates are given in Table 3. This is essential, since the net lateral volume fluxes in and out of the Arctic are very small ($\sim$0.2 Sv in the annual mean) compared to the fluxes through individual straits (e.g. $\sim$2.3 Sv for Davis Strait, Curry et al., 2011).

## 3.2 Variational approach for budget closure

Given all the various, largely independent data sets we use, closure between the budget terms will not be perfect, resulting in a budget residual. To get rid of any residual and obtain a closed budget with physical terms only, we follow methods by Mayer et al. (2019), L'Ecuyer et al. (2015) and Rodell et al. (2015) and use a variational Lagrange multiplier approach to enforce budget closure on annual and monthly scales. Therefore the following cost function J is minimized:

$$J = \sum_i \frac{(F_i - F'_i)^2}{\sigma'^2_i} + \lambda \sum_i F'_i \qquad (6)$$

With the Lagrange multiplier $\lambda$, the a priori estimates of the budget terms $F'_i$, the adjusted budget terms $F_i$, the uncertainty of the respective budget term $\sigma'^2_i$ and the budget residual $\sum_i F'_i$.

*Annual optimization*

Inserting the annual means of the individual budget terms into Eq. (6) and differentiation in respect to $\lambda$ and the a priori estimates of the budget terms, yields eight equations with eight unknowns. Solving the system of equations results in an expression for the adjusted budget terms $F_k$:

$$F_k = F'_k + \frac{\sigma'^2_k}{\sum_i \sigma'^2_i} \sum_i F'_i \qquad (7)$$

Hence, the budget residual is distributed across the budget terms according to their relative uncertainty. The a priori uncertainties are derived from the standard deviations of the mean annual budget terms. The a posteriori uncertainties $\sigma^2_k$ are calculated following Mayer et al. (2018):

$$\sigma^2_k = \left( \frac{1}{\sigma'^2_k} + \frac{1}{\sum_i \sigma'^2_i - \sigma'^2_k} \right) \qquad (8)$$

*Monthly optimization*

Monthly optimization is performed in two steps. First adjusted fluxes are calculated for each month separately following Eq. (7), whereat the a priori uncertainty is estimated by taking the maximum of the seasonal standard deviations and is kept fixed throughout all months. However, the annual means of the resulting monthly fluxes do not coincide with the annually optimized

fluxes. Therefore we follow Rodell et al. (2015) and apply a second Lagrangian optimization, where the adjusted monthly fluxes from the first step ($F_k$) are adjusted in relation to their uncertainty, so that their annual mean is equal to the annually optimized fluxes $F_m$:

$$FO_k = F_k + \frac{12\sigma_k'^2}{\sum_i \sigma_i'^2}\left(F_m - \frac{1}{12}\sum_i F_i\right) \tag{9}$$

However, the second step again generates small monthly residuals. Therefore the whole procedure is performed iteratively a second time, using the a posteriori uncertainties gained through Eq. (8). This results in the desired monthly fluxes, that satisfy both, a closed budget and consistency with the annually optimized fluxes.

### 3.3 Trend and relative error calculation

We calculate relative, decadal trends following Zsótér et al. (2020) and Stahl et al. (2012) by applying a linear regression to the
annual mean time series:

$$trend = \frac{10 * slope}{mean} \tag{10}$$

The *slope* of the time series is the annual change obtained through the linear regression and the *mean* is the long-term annual mean of the timeseries. The multiplication factor 10 results as we calculate trends over a fixed 10-year period. Hence, a trend of e.g. 0.1 is equal to an increase of 10% over a decade. All trends are calculated over the common period of the discharge
datasets 1981-2019, except for GloFAS$_{E5L}$ which is calculated over 1999-2018. We do not consider temporal auto-correlation, assuming that subsequent annual means are only weakly correlated, and determine significance using the Wald Test with a t-distribution, where p-values smaller than 0.05 are considered as significant.

To compare river discharge estimates from the various reanalyses to river discharge observations we use the Pearson's correlation coefficient r and a normalised root mean square error (NRMSE), which is calculated by dividing the RMSE through the
RMS of the observed values ($NRMSE(x)=RMS(x-obs)/RMS(obs)$).

## 4  Results and Discussion

We first discuss seasonal cycles and trends for the four major Arctic catchments - Yenisei, Ob, Lena and Mackenzie. Then we extend our assessments to the Pan-Arctic region, where we compare the total terrestrial Arctic runoff with oceanic volume fluxes through the main gateways.

### 4.1  Analysis of major catchments

### 4.1.1  Seasonal cycles

Figure 2 shows seasonal cycles of various hydrological components for the Yenisei, Ob, Lena and Mackenzie catchments. The top panels compare runoff from ERA5 and ERA5-Land with river discharge from GLOFAS$_{E5}$, GLOFAS$_{E5L}$ and GLOFAS$_{E5_{new}}$,

as well as with observed discharge values.

Observations show a distinct runoff peak in June due to snow melt and river ice breakup, and weak runoff through winter. The spring flood season of Eurasian rivers depends on the basin size and usually ends by the end of June at small size rivers and by the end of July or beginning of August at large rivers like Ob, Yenisei and Lena (Yang et al., 2007). While smaller rivers usually exhibit a low-flow season in the summer to fall period, discharge from larger rivers mostly shows a slower decrease, also because of summer rainfalls providing additional discharge water. Especially at rivers of Eastern Siberia (e.g. Lena) intense

rainfalls in the summer-fall season may occasionally cause rainfall floods (Shiklomanov et al., 2021a). In late winter, with the maximum of river freeze-up, discharge reaches its minimum.

Runoff data from ERA5 and consequently also from GLOFAS$_{E5}$ underestimate the summer peaks recorded by gauges and reach only about 25 to 50% of the observed peak discharge values. In the low flow season the reanalyses slightly exceed observed discharge, however this does not alter the annual means considerably resulting in a clear underestimation of discharge

by ERA5 and GLOFAS$_{E5}$ also in annual terms. The difference between ERA5 runoff and GLOFAS$_{E5}$ discharge is expected to be caused by two sink terms, a groundwater loss component, calibrated in LISFLOOD, that removes water that is lost to deep groundwater systems, and the open water evaporation component, which also removes water through evaporation over water surfaces in LISFLOOD. The negative contribution can reach up to 20-40% of the average flow by any of the two terms in certain regions (described in the LISFLOOD model documentation, Burek et al., 2013). Better accordance in terms of peak height

and annual means is achieved by runoff from ERA5-Land - except for the Lena basin - while the sink terms in GloFAS again causes an underestimation by GLOFAS$_{E5L}$. At the Ob basin the runoff peak in ERA5-Land, and also ERA5, occurs in May, thus a month earlier than the observed peak. GloFAS discharge is not in phase with ERA5 runoff, but reaches its peak in June, presumably due to the delay by the river routing component. In contrast to GLOFAS$_{E5}$ and GLOFAS$_{E5L}$, GLOFAS$_{E5_{new}}$ reaches values similar to observations and agrees best with the observed values in terms of annual means, peak heights and

seasonality. The middle panels of Fig. 2 show snowfall and snow melt as well as the atmospheric components net precipitation (P-E) and divergence of moisture flux (VIWVD). Seasonal cycles of P-E and VIWVD minus storage change agree quite well in terms of peak heights and timing, with low moisture inflow and low net precipitation in summer and higher values in autumn and winter. Annual values of P-E and VIWVD-$\Delta$S differ by 2-16% depending on the catchment. Seasonal cycles of ERA5 snow melt show that there is a lag of one month between the peak in snow melt and river discharge. This can partly be explained

by the time it takes for the water to reach the river mouth and by water resource management effects, but is mostly caused due to delayed river ice break up in the lower parts of the basins. For example, at the upper part of the Ob river, ice breaks up around April to May, while the lower part breaks up between May and June (Yang et al., 2004b). While human impacts through water withdrawals for agricultural use are rather limited compared to rivers in lower latitudes, water resource management via dams and reservoirs can significantly alter the seasonal discharge cycle of the larger Arctic rivers (Shiklomanov et al., 2021a). Espe-

cially the Ob and Yenisei basins, but also Lena and Kolyma, are affected by multiple reservoirs using water for hydroelectric power generation and delaying discharge from high-flow periods to the low-flow season (Lammers et al., 2001; Ye et al., 2003; Yang et al., 2004b, a; Shiklomanov et al., 2021a).

The lower panels of Fig. 2 show land storage change from ERA5 and from GRACE. Additionally, ERA5 storage is separated

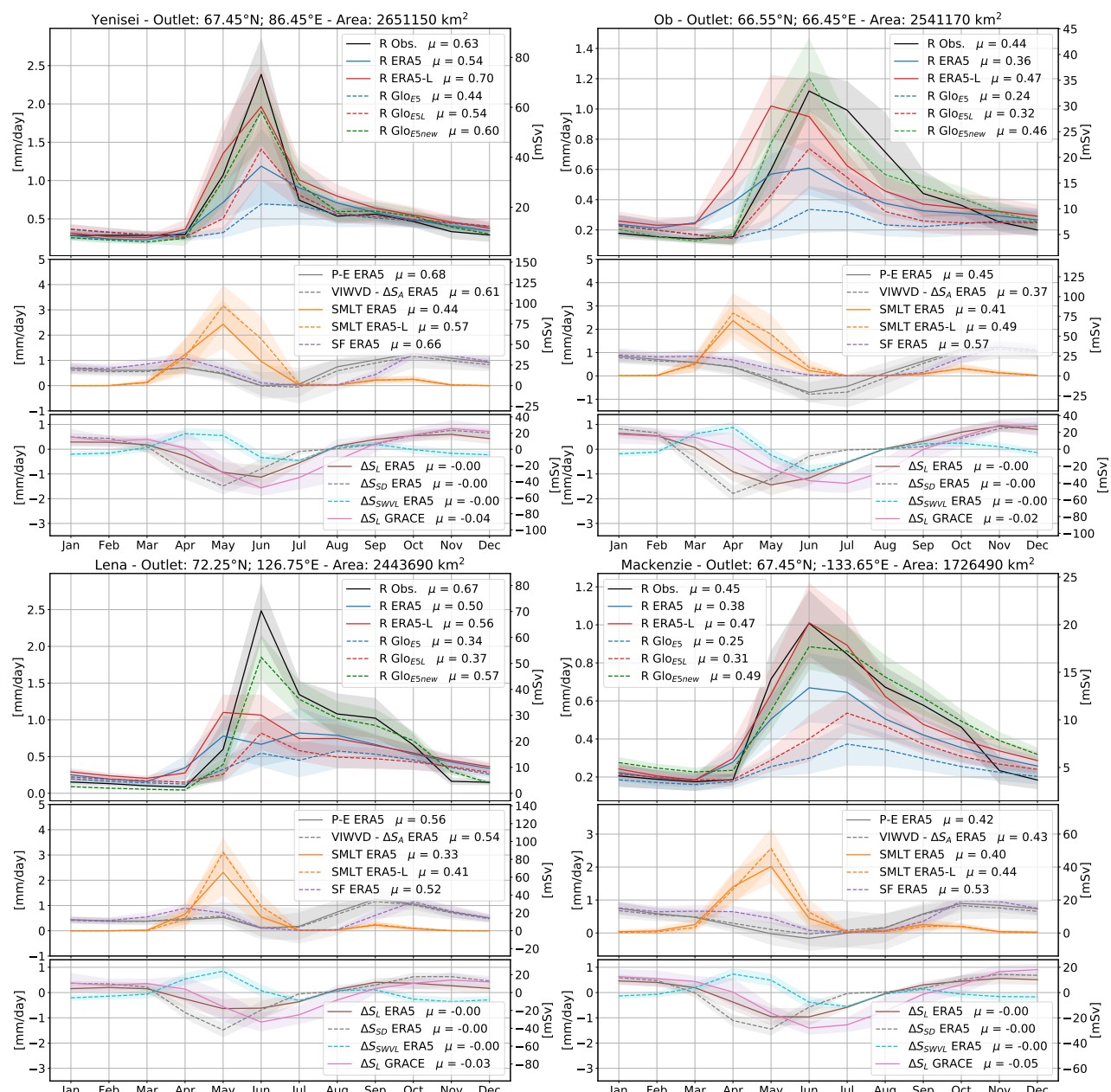

**Figure 2.** Yenisei, Ob, Lena and Mackenzie seasonal cycles of runoff and river discharge (R) in the top panel, terrestrial net precipitation (P-E), divergence of moisture flux (VIWVD), snow melt (SMLT) and snowfall (SF) in the middle and land storage change ($\Delta S_L$), snow depth change ($\Delta S_{SD}$) and soil water change ($\Delta S_{SWVL}$) in the bottom panel. Seasonal cycles are calculated over the 1981-2019 period (1999-2018 for GloFAS$_{E5L}$). Shading denotes the ± one standard deviation, $\mu$ are the long-term means. Furthermore, geographic coordinaets of gauge observations and catchment areas are given. Long term annual means in mm/day are given in the legend.

into its components of soil water change $\Delta S_{SWVL}$ and snow depth change $\Delta S_{SD}$. In autumn and winter, land water is accumulated through a rise in snow depth, in summer storage is lost through snow melt and associated runoff, and to a smaller part also through evaporation. GRACE shows a significant annual loss of water storage over the past decades, while storage change in ERA5 exhibits only a slight annual decline. In terms of seasonality GRACE features the largest storage changes in June and July, while $\Delta S$ from ERA5 tends to peak 1-2 month earlier. Again this could be caused by delayed river ice breakup and backwater that is observed by GRACE, but not represented in ERA5.

### 4.1.2 Trends

Figure 3 shows annual river discharge values for Yenisei, Ob, Lena and Mackenzie rivers. The corresponding temporal means, standard deviations, decadal trends, normalised RMSEs and correlation coefficients are given in Tables 4 and 5. Observations indicate a slight increase in river discharge over the past decades. The largest changes occurred at the Ob and Lena basins with a rise of about 4% per decade. Yenisei shows no significant long term trend, as, after a rise until the early 2000s, discharge values seem to have slightly decreased over the past decade. In contrast to observations, runoff from ERA5 and discharge from GLOFAS$_{E5}$ show distinct negative trends of 11 to 16% per decade. These strong decreases are a result of changing biases in the snow assimilation in ERA5 and are discussed further in Sect. 3.3.3. The effect of the sink terms removing water in GLOFAS$_{E5}$, and hence producing a negative shift, results in a strong underestimation towards the end of the time series with values reaching only slightly more than 50 % of the observed values. Runoff from ERA5-Land shows a distinct improvement, with long term means generally deviating only about 5% from observations (10% for Lena) and reasonable NRMSE values. However, the sink terms in GLOFAS$_{E5L}$ again lead to an underestimation of the observed discharge values. Calculating runoff indirectly through net precipitation minus land storage change and VIWVD minus atmospheric storage change (Eq. (3)) yield results mostly within 10% of the observed discharge values.

In contrast to GLOFAS$_{E5}$ and GLOFAS$_{E5L}$, GLOFAS$_{E5_{new}}$ exhibits greatly improved river discharge values and concerning the normalised RMSE values it provides the best results when compared to observations. Both discharge from GLOFAS$_{E5_{new}}$ and estimation through P-E also feature slightly negative trends, only calculation through VIWVD exhibits no, or even slightly positive trends.

### 4.2 Pan-Arctic approach

To get a complete estimate of the total amount of freshwater entering the Arctic Ocean via land, which is needed for subsequent budget calculation, river discharge is calculated over the whole Pan-Arctic area. First we look into the Pan-Arctic seasonal cycle and afterwards at annual means and long-term trends over the whole Arctic drainage area.

Pan-Arctic freshwater discharge estimates vary significantly between different studies. This comes to a big part from different definitions of the geographic area (Prowse and Flegg, 2000; Shiklomanov and Shiklomanov, 2003) as there is no strict boundary to the south that defines the Arctic and past studies disagree whether to include e.g. Greenland and the Hudson Bay or not. Another reason for discrepancies between different studies comes due to different approaches of discharge evaluation (Shiklomanov et al., 2021a) - some approaches are based on model simulations and others are derived from river discharge

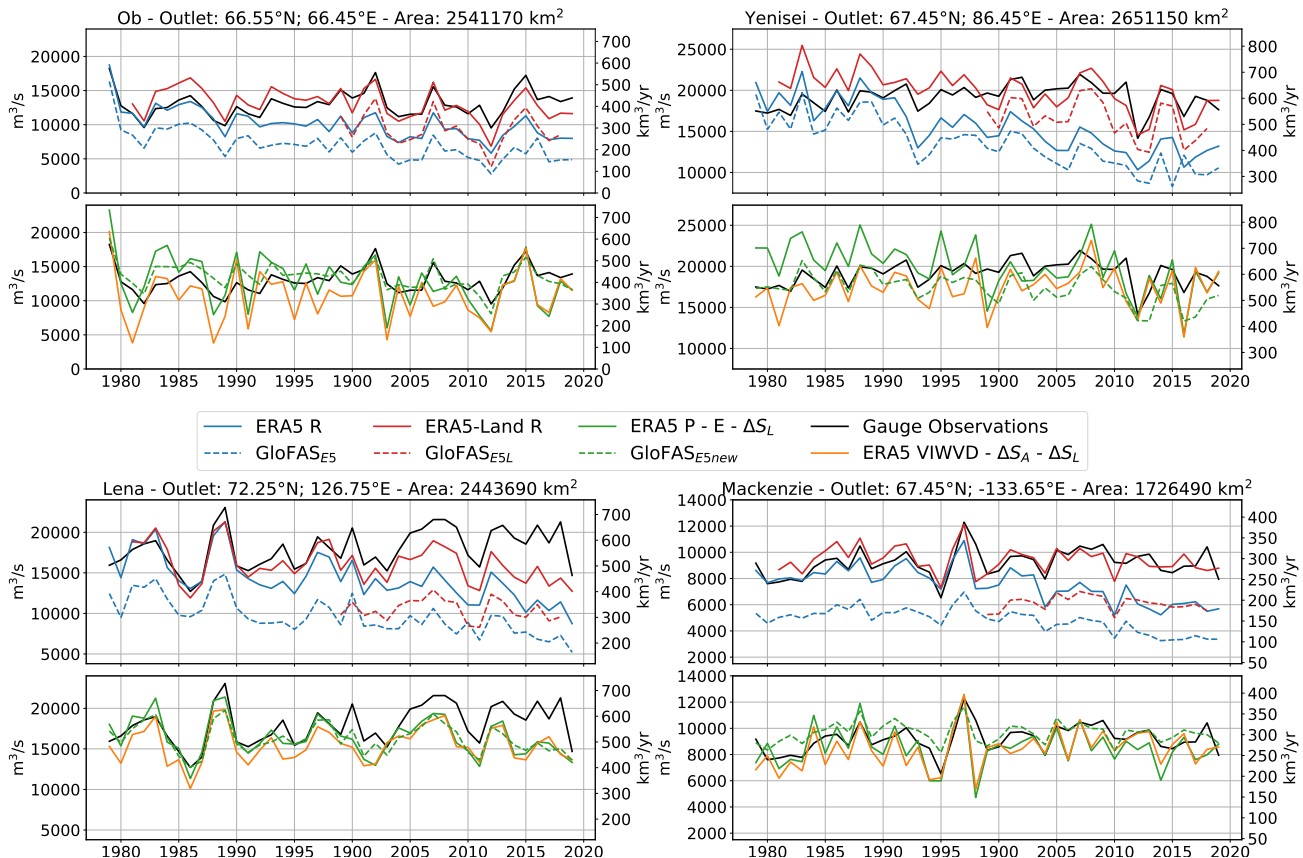

**Figure 3.** Annual means of observed river discharge, ERA5 runoff, ERA5 net precipitation, ERA5 VIWVD and ERA5-Land runoff as well as $GLOFAS_{E5}$, $GLOFAS_{E5L}$ and $GLOFAS_{E5new}$ for the period of 1979 to 2019 (1981-2019 for ERA5-Land and 1999-2018 for $GLOFAS_{E5L}$).

measurements. Using only in-situ measurements poses several challenges - especially the handling of the large unmonitored areas, that account for about 30–40% of the total drainage area (Shiklomanov et al., 2002). Most studies adopt the method of hydrological analogy (Shiklomanov and Shiklomanov, 2003), a method where total discharge is calculated by extrapolat-

ing gauged runoff to the unmonitored rivers and streams. However, hydrological, climatic and land cover conditions between gauged and ungauged areas can differ quite a lot, resulting in inaccurate estimates (Shiklomanov et al., 2021a). Hence, we compare two different estimation methods for observed discharge that are displayed in Fig. 4. For that purpose we consider gauging records from the 24 largest Arctic rivers, that amount for roughly 70% of the total drainage area used in this study. Pan-Arctic river discharge for the ungauged areas for each month was estimated as follows: Firstly following e.g. Serreze et al.

(2006) and applying the method of hydrological analogy for each calendar month by transforming discharge into runoff and, with the assumption that runoff from the ungauged area is the same as runoff from the gauged area, multiplication with the

|  |  | Gauge Observations | units |
|---|---|---|---|
|  | $\mu$ | 19.2 | $[10^3 m^3 s^{-1}]$ |
| Yenisei | $\sigma$ | 1.5 | $[10^3 m^3 s^{-1}]$ |
|  | Trend | $0.00\pm0.01^n$ | [fraction/decade] |
|  | $\mu$ | 12.9 | $[10^3 m^3 s^{-1}]$ |
| Ob | $\sigma$ | 1.8 | $[10^3 m^3 s^{-1}]$ |
|  | Trend | $0.04\pm0.02$ | [fraction/decade] |
|  | $\mu$ | 18.0 | $[10^3 m^3 s^{-1}]$ |
| Lena | $\sigma$ | 2.5 | $[10^3 m^3 s^{-1}]$ |
|  | Trend | $0.04\pm0.02$ | [fraction/decade] |
|  | $\mu$ | 9.2 | $[10^3 m^3 s^{-1}]$ |
| Mackenzie | $\sigma$ | 1.0 | $[10^3 m^3 s^{-1}]$ |
|  | Trend | $0.02\pm0.02$ | [fraction/decade] |

**Table 4.** Mean values and standard deviations of observed river discharge as well as relative decadal trends and their standard errors (see Eq. (10)), calculated over the period 1981-2019. Trends that are not significant are marked with an superscript [n].

whole drainage area to transform back to river discharge - here denoted as *ae* (area estimate). However, taking runoff from reanalysis (e.g. ERA5-Land) for the same 24 drainage areas and expanding it over the whole area yields different results, with smaller summer peaks, than taking runoff directly from the total drainage area. Hence, monthly correction factors are calculated from our most reliable products ERA5-Land and GloFAS$_{E5_{new}}$ and by multiplication with those, a more accurate estimate of observed Pan-Arctic river discharge should be possible - denoted as *Ee* (ERA estimate) and *Ge* (GloFAS estimate), leading to more plausible high flow peaks. Following the results of the previous sections we trust most in GloFAS$_{E5_{new}}$ as it featured the smallest NRMSE values and highest correlation coefficients and hence use the GloFAS estimate *Ge* for the following Pan-Arctic analysis.

### 4.2.1 Seasonal Cycle

Figure 5 shows the mean seasonal cycles of the various Pan-Arctic river discharge estimates for the period 1981 to 2019 (1999 to 2018 for GLOFAS$_{E5L}$).

Results are similar as for the individual catchments, with the largest amounts of water entering the Arctic Ocean in June. Observations (*Ge*) show summer peaks of about $4.8\times10^5$ m$^3$s$^{-1}$ and annual means of 4031 km$^3$yr$^{-1}$. ERA5 runoff and discharge from GLOFAS$_{E5}$ and GLOFAS$_{E5L}$ clearly underestimate the summer peaks and annual means, however they slightly overestimate runoff in the low-flow winter months. ERA5-Land runoff performs quite well in terms of annual means, but the June maximum is still about 25-35% too low. GLOFAS$_{E5_{new}}$ performs best in terms of seasonality, but still slightly underestimates the summer peak. Indirectly calculated discharge through Eq. (3) does not reach those distinct high June peaks. Annual values from calculations through P-E reach values similar to ERA5-Land runoff, while the VIWVD estimate is about 5% too low. The

| | | E5 R | E5 P-E | E5 VIWVD | E5-L R | GloFAS$_{E5}$ | GloFAS$_{E5L}$ | GloFAS$_{E5new}$ |
|---|---|---|---|---|---|---|---|---|
| | $\mu$ | 15.5 | 19.8 | 17.4 | 20.0 | 13.4 | 16.4* | 17.3 |
| | $\sigma$ | 3.0 | 2.9 | 2.4 | 2.4 | 2.9 | 2.3* | 1.9 |
| Yenisei | Trend | -0.14±0.02 | -0.06±0.02 | 0.01±0.02$^n$ | -0.06±0.01 | -0.16±0.02 | -0.10±0.05*$^n$ | -0.05±0.01 |
| | NRMSE | 0.25 | 0.15 | 0.14 | 0.11 | 0.34 | 0.17* | **0.10** |
| | r | 0.24 | 0.37 | 0.49 | 0.53 | 0.21 | 0.81* | **0.64** |
| | $\mu$ | 9.9 | 12.6 | 10.6 | 12.5 | 6.8 | 9.3* | 13.4 |
| | $\sigma$ | 1.8 | 2.9 | 3.3 | 2.1 | 1.7 | 2.3* | 1.7 |
| Ob | Trend | -0.11±0.02 | -0.08±0.03 | 0.02±0.04$^n$ | -0.06±0.02 | -0.15±0.03 | -0.10±0.99*$^n$ | -0.03±0.02$^n$ |
| | NRMSE | 0.28 | 0.21 | 0.25 | 0.13 | 0.45 | 0.32* | **0.11** |
| | r | 0.39 | 0.40 | 0.58 | 0.66 | 0.32 | 0.89* | **0.68** |
| | $\mu$ | 14.2 | 16.4 | 15.6 | 16.1 | 9.6 | 10.5* | 16.2 |
| | $\sigma$ | 2.8 | 2.4 | 2.2 | 2.3 | 2.2 | 1.3* | 1.8 |
| Lena | Trend | -0.12±0.02 | -0.03±0.02$^n$ | 0.00±0.02 | -0.05±0.02 | -0.15±0.02 | -0.03±0.05*$^n$ | -0.02±0.02$^n$ |
| | NRMSE | 0.27 | 0.14 | 0.17 | 0.15 | 0.50 | 0.45* | **0.14** |
| | r | 0.33 | 0.65 | 0.74 | 0.66 | 0.27 | 0.73* | **0.76** |
| | $\mu$ | 7.5 | 8.6 | 8.5 | 9.4 | 4.8 | 6.1* | 9.7 |
| | $\sigma$ | 1.3 | 1.5 | 1.5 | 1.0 | 0.9 | 0.5* | 0.7 |
| Mackenzie | Trend | -0.11±0.02 | -0.01±0.03$^n$ | 0.03±0.02 | -0.02±0.01$^n$ | -0.13±0.02 | -0.00±0.04*$^n$ | -0.01±0.01$^n$ |
| | NRMSE | 0.24 | 0.17 | 0.16 | 0.09 | 0.49 | 0.36* | **0.09** |
| | r | 0.29 | 0.49 | 0.55 | 0.65 | 0.35 | 0.65* | **0.73** |

**Table 5.** Mean values and standard deviations of river discharge as well as relative decadal trends and their standard errors (see Eq. (10)), calculated over the period 1981-2019 (1999-2018 for GloFAS$_{E5L}$, indicated by the superscript *). Units are the same as in Table 4. Trends that are not significant are marked with an superscript $^n$. Additionally, normalised RMSE values and correlation coefficients for annual means, calculated in relation to observations (Table 4), are given. Bold values identify the best correlation coefficients and NRMSE values.

runoff climatology Bt06 exhibits a similar seasonal cycle as our observation based estimate, however it is roughly 9% lower than our estimate.

The bottom panel of Fig. 5 shows land storage change components. The time lag between the observed storage change through GRACE and ERA5 is clearly evident again. And just as for the four major basins, also for the sum of all Eurasian and North American catchments (excluding Greenland) GRACE data show a major decline of land water storage over the past decades,

reaching -131 km$^3$ per year for our area of interest, while land storage from ERA5 shows considerably smaller declines of -34 km$^3$ per year. Additionally, GRACE water storage shows a strong decline of -115 km$^3$ per year over the northwestern part of the Greenlandic ice cap (as specified in Fig. 1), raising the total Pan-Arctic storage change to 266 km$^3$ per year. The largest changes for GRACE water storage occur over the coastal areas of Greenland and the Canadian Arctic Archipelago, further declining trends can be seen over the Arctic islands Svalbard and Novaya Zemlya, as well as over mountainous regions (see

Fig. A1 in the appendix). This suggests a tight linkage to glacial melting.

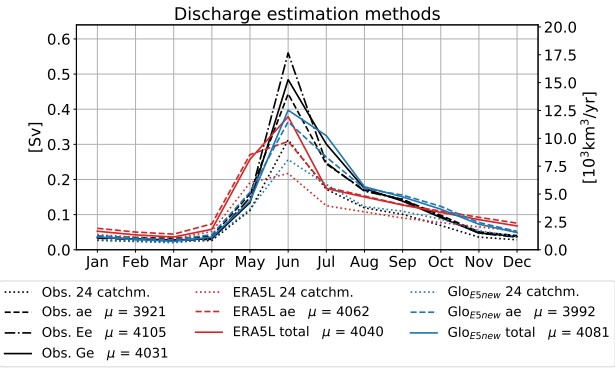

**Figure 4.** Pan-Arctic discharge estimates using hydrological analogy (ae) and monthly correction factors from ERA5-Land (Ee) and GloFAS$_{E5_{new}}$ (Ge) for the period of 1981-1999. Long term annual means in km$^3$yr$^{-1}$ are given in the legend.

Wouters et al. (2019) use monthly GRACE Stokes coefficients to examine global glacier mass losses for regions defined in the Randolph Glacier Inventory (RGI), excluding Greenland. Computing the sum over all regions that roughly resemble our area of interest, yields a glacial mass change of about -100 Gt (gigatons), or roughly -109 km$^3$ per year. Hence, the sum of glacial mass change and storage change from ERA5 resembles the land storage change from GRACE within 10%.

### 4.2.2 Trends

Long term means of annual discharge over the coinciding period of all datasets 1999-2018 (1981-2018 in brackets) and the distribution to Eurasia, North America and the CAA can be seen on the left panel of Fig. 6 and long term annual means, relative trends, correlation coefficients and NRMSE values are provided in Table 6. About 75 % of Pan-Arctic runoff come from the Eurasian watersheds, followed by North America with about 17 % and the smallest contribution coming from the CAA. Total Pan-Arctic runoff values, calculated over 1999-2018, range between 2625 km$^3$yr$^{-1}$ for GLOFAS$_{E5}$ and 3952 km$^3$yr$^{-1}$ for indirectly calculated runoff through P-E-$\Delta S_L$ from ERA5. Our observation-based estimates are considerably higher, reaching 4117 km$^3$yr$^{-1}$. Using observations of river discharge only and omitting the CAA, Serreze et al. (2006) obtain values of 3200 km$^3$yr$^{-1}$. Including the CAA, but excluding Yukon river, Haine et al. (2015) combine runoff from ERA-Interim with river discharge observations and report annual values of 3900 $\pm$ 390 km$^3$yr$^{-1}$ for the period 1980-2000 and 4200 $\pm$ 420 km$^3$yr$^{-1}$ for the period of 2000-2010. Our observation-based estimates, including Yukon river, reach values of 3971 $\pm$ 31 km$^3$yr$^{-1}$ for 1980-2000 and 4155 $\pm$ 42 km$^3$yr$^{-1}$ for 2000-2010. Subtracting the contribution of Yukon River (about 200 km$^3$yr$^{-1}$), our estimates are about 5% lower than those from Haine et al. (2015). Excluding Yukon river from ERA5-Land runoff and indirectly calculated runoff through ERA5 P-E, both estimates are quite close to the estimates made by Haine et al. (2015) for the period of 1981-2000, however they are substantially too low in the 2000-2010 period.

For the whole Pan-Arctic region, both ERA5 and GLOFAS$_{E5}$ feature runoff decreases of 10-11 % per decade, and ERA5-Land and indirectly calculated runoff through ERA5 P-E exhibit negative trends of about 2-3 % per decade. In contrast, the runoff

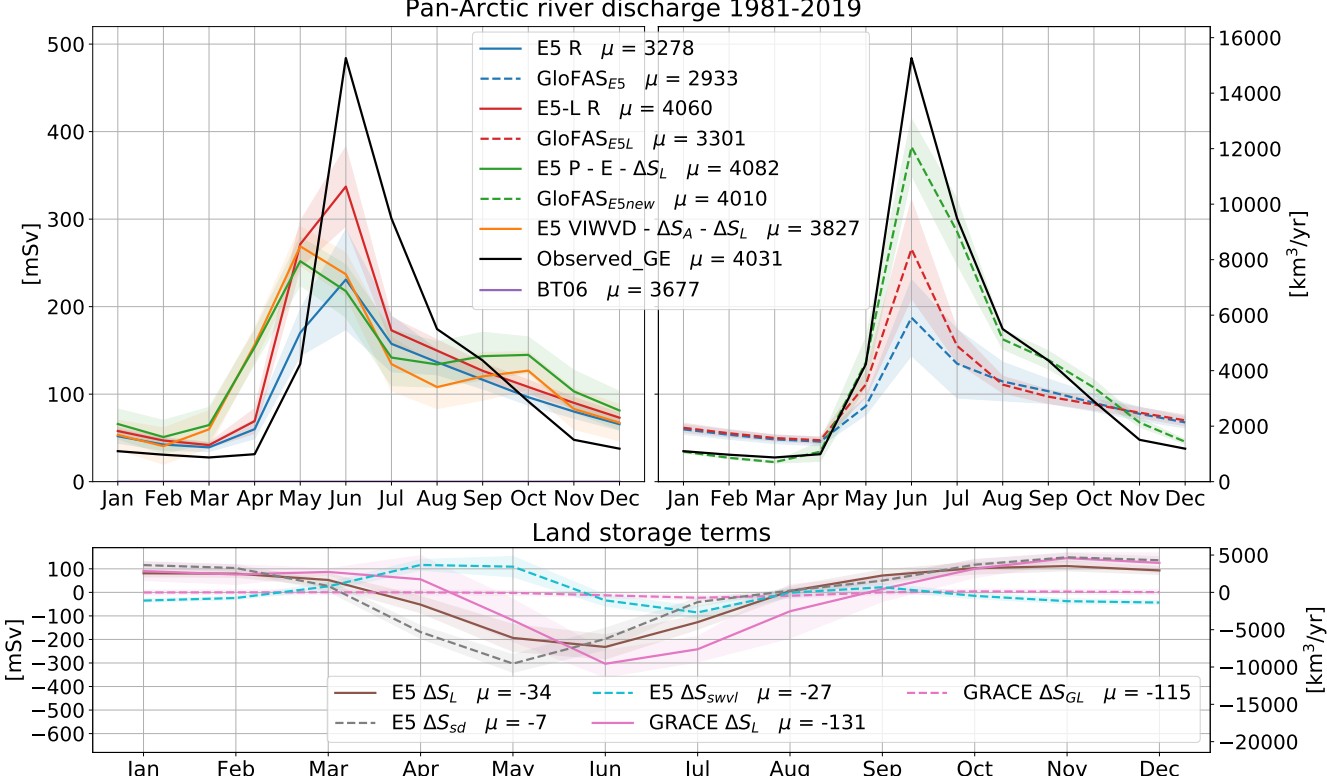

**Figure 5.** Top panel: Seasonal Cycles of Pan-Arctic runoff for ERA5, ERA5-Land and various GloFAS versions, as well as indirectly calculated runoff through P-E and VIWVD. Additionally, an estimate of observed river discharge is given. Seasonal Cycles are calculated over the period 1981-2019 (1999-2018 for GLOFAS$_{E5L}$). Bottom panel: storage change components from GRACE and ERA5. ERA5 snow-depth (dashed brown) and soil water change (dashed cyan) sum up to the total ERA5 storage change (brown solid). Shading denotes variance as the $\pm$ one standard deviation. Furthermore, long term annual means in km$^3$yr$^{-1}$ are given and appropriately the right axis is scaled in km$^3$yr$^{-1}$.

estimate through ERA5's VIWVD features a slight increase of 2%, identically to the trend present in our observation-based estimate. GloFAS$_{E5_{new}}$ comes closest to gauge observations concerning annual means and NRMSE values, while indirectly calculated runoff through VIWVD features the same trend as observations and also the highest correlation concerning annual means. However, the VIWVD estimate generally yields a roughly 5% lower discharge compared to observations.

The strong decreases in ERA5 runoff are reinforced through two discontinuities in the dataset, one around 1992 and the second one around 2004, that lead to a significant drop and hence a clear underestimation of runoff over the past decades. As GLOFAS$_{E5}$ takes ERA5 as input, it also adopts those discontinuities. In contrast, ERA5-Land does not exhibit such breaks, suggesting that the error may come from the data assimilation system in ERA5. While the discontinuity around 2004 was traced back to the introduction of the IMS (Interactive Multisensor Snow and Ice Mapping System) snow product into the

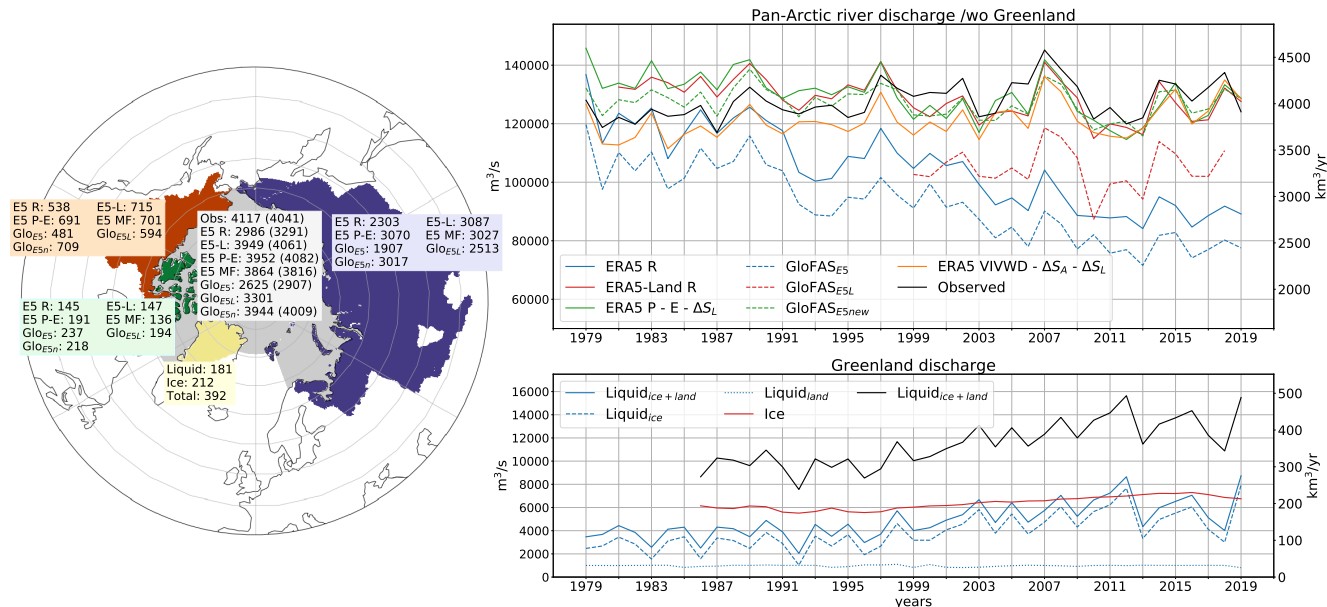

**Figure 6.** The left panel shows the drainage area and mean annual discharge values in $\text{km}^3\text{yr}^{-1}$ for the whole area, as well as for Eurasia, North America and the CAA individually. Values are calculated over the coinciding period of all datasets 1999-2018 (values in brackets are calculated over the period 1981-2019). The observation based estimate is calculated using correction factors for the ungauged areas from $\text{GLOFAS}_{E5_{new}}$. On the right hand side annual means of our observation based estimate, ERA5, ERA5-Land, $\text{GLOFAS}_{E5}$, $\text{GLOFAS}_{E5L}$ and $\text{GLOFAS}_{E5_{new}}$ runoff, as well as indirectly calculated runoff through P-E and VIWVD for the period 1979-2019 (1981-2019 for ERA5-Land and 1999-2018 for $\text{GLOFAS}_{E5L}$) are shown. Additionally, Greenlandic liquid (from land and ice; 1979-2019) and solid (1986-2019) discharges are displayed in the bottom right panel.

assimilation system (Zsótér et al., 2020), the reason for the break around 1992 could not be identified yet and is discussed further in the following section.

Contrary to discharge estimates from reanalyses, observations show opposing trends and a rise in river discharge over the past decades. In NOAA's Arctic Report Card 2018 Holmes et al. (2018) stated that river discharge from Eurasians largest rivers
(Ob, Yenisei, Lena, Kolyma, Pechora, Severnaya Dvina) has increased by $3.3 \pm 1.6$ % per decade since 1976 and by $2.0 \pm 1.8$ % per decade for the largest North American rivers (Mackenzie and Yukon). Our observation-based estimates show an increase of $2 \pm 1\%$ for the whole Pan-Arctic region.

Runoff from Greenland was left out so far, but contributions of liquid and ice discharge are not negligible. Additionally to Eurasian and North American discharge, Fig. 6 shows discharge from the Greenlandic ice and land areas north of our
boundaries in Davis and Fram Strait. Greenlandic liquid discharge from land and ice is taken from Mankoff et al. (2020a), who calculate discharge from daily runoff estimates of the Modèle Atmosphérique Régional (MAR) and the Regional Atmospheric Climate Model (RACMO). We estimate the optimal discharge by taking the mean of MAR and RACMO discharge. Solid ice discharge through calving of marine-terminating glaciers and melt-water from ice–ocean boundary melting at submarine

| | Observed $Ge$ | E5 R | E5 (P-E) | E5 VIWVD | E5-L R | GloFAS$_{E5}$ | GloFAS$_{E5L}$ | GloFAS$_{E5new}$ | units |
|---|---|---|---|---|---|---|---|---|---|
| $\mu$ | 127.8 | 104.4 | 129.4 | 121.0 | 128.8 | 92.2 | 104.7* | **127.1** | $[10^3 m^3 s^{-1}]$ |
| $\sigma$ | 6.1 | 13.0 | 7.1 | 5.9 | 6.6 | 11.3 | 7.0* | 5.1 | $[10^3 m^3 s^{-1}]$ |
| Trend | 0.02±0.01 | -0.10±0.01 | -0.03±0.01 | **0.02±0.01** | -0.02±0.01 | -0.11±0.01 | 0.00±0.03*$^n$ | -0.01±0.01 | [fraction/decade] |
| NRMSE | - | 0.22 | 0.06 | 0.06 | 0.06 | 0.30 | 0.20* | **0.04** | |
| r | - | -0.14 | 0.29 | **0.78** | 0.38 | -0.15 | 0.85* | 0.53 | |

**Table 6.** Mean values and standard deviations of Pan-Arctic river discharge in $m^3 s^{-1} \times 10^{-3}$ as well as decadal relative trends, calculated over the period 1981-2019 (1999-2018 for GLOFAS$_{E5L}$, indicated by *). Insignificant trends are indicated by the superscript $^n$. Bold values identify the best estimates in respect of long term means, trends, correlation coefficients, NRMSE values.

| | | Liquid$_{Ice}$ | Liquid$_{Land}$ | Liquid$_{total}$ | Ice | Total | units |
|---|---|---|---|---|---|---|---|
| | $\mu$ | 4.2 | 1.0 | 5.2 | 6.7 | 11.9 | $[10^3 m^3 s^{-1}]$ |
| Greenland | $\sigma$ | 1.5 | 0.1 | 1.5 | 0.5 | 2.0 | $[10^3 m^3 s^{-1}]$ |
| | Trend | 0.25±0.05 | 0.01±0.02$^n$ | 0.20±0.04 | 0.08±0.01 | 0.13±0.02 | [fraction/decade] |

**Table 7.** Mean values, standard deviations and trends for liquid discharge from ice and land as well as solid ice discharge for Greenland north of Davis and Fram Strait over the period 1987-2019. Insignificant trends are indicated by the superscript $^n$.

glaciers is taken from Mankoff et al. (2020b) over the regions central west (CW), north west (NW) and north (NO) (see Mankoff et al. (2020b), their Fig. 1) to roughly account for our region of interest. With a total contribution of 392 km$^3$yr$^{-1}$ over the period 1999-2018, discharge from Greenland accounts for roughly 10% of total Pan-Arctic discharge. Most of the freshwater supplied to the Arctic Ocean comes from solid ice discharge, followed by liquid discharge from glaciers and only a small contribution from land runoff. Liquid discharge from land and ice show pronounced seasonalities with peaks in June and July, while solid ice discharge stays relatively stable throughout the year. The bottom figure of the right panel of Fig. 6 shows annual means for the individual Greenlandic discharge components and Table 7 displays mean values and trends. Liquid ice discharge exhibits a vast positive trend of 26% per decade and also solid ice discharge shows a clear rise of about 8% per decade. It can be expected that Greenland discharge will further increase in the future (Muntjewerf et al., 2020; Church et al., 2013; Vaughan et al., 2013, e.g.)

Adding Greenlandic discharge to our $Ge$ observation estimate yields a total (pan-Arctic plus Greenland) discharge of 4423 km$^3$yr$^{-1}$. A recent assessment (Shiklomanov and Lammers, 2013; Shiklomanov et al., 2021a) estimates the total discharge to the Arctic Ocean at approximately 4300 km$^3$yr$^{-1}$ for the period 1936-2006. Differences may stem from the use of different data products, from slight variations in discharge area - Shiklomanov and Lammers (2013) use an area of approximately $19 \times 10^6$ km$^2$, while our area together with the considered Greenlandic area ($0.95 \times 10^6$ km$^2$) sums up to $19.15 \times 10^6$ km$^2$ - and from different calculation periods - discharge was very likely smaller in the mid-20th century.

### 4.2.3 ERA5 Runoff discontinuities

As mentioned above, a possible reason for the negative shifts in ERA5 runoff lies in the data assimilation system and its removal of soil moisture. Zsótér et al. (2020) assess the GLOFAS$_{E5}$ river discharge reanalysis as well as ERA5 and ERA5-Land runoff and compare them with available river discharge observations. Similarly to our diagnostics for Arctic rivers, Zsótér et al. (2020) find river discharge decreases in GloFAS$_{E5}$ for several rivers of the world, that are not supported by observations. They find trends in tropical and subtropical areas being driven by changes in precipitation, while changes in snow melt have a very strong influence on river discharge trends in the northern latitudes. Thus, the runoff decreases in the northern high latitudes are likely linked to snow assimilation and other processes related to snow melt. Figure 7 shows ERA5 snowfall, snow melt and the sum of snow melt and snow evaporation as well as the corresponding parameters from ERA5-Land for the whole Pan-Arctic region, Eurasia, North America and the CAA. ERA5 shows substantial differences between snow gain and snow loss, and Pan-Arctic snow melt exhibits the same discontinuities around 1992 and 2004 as runoff from ERA5. In contrast, snow gain and snow loss are in balance for ERA5-Land. Zsótér et al. (2020) find similar results and attribute the differences to the land data assimilation, that has impacts on snow and soil moisture. The discontinuity in 2004 was traced back to a change in operational snow analysis, through the introduction of the 24-km Interactive Multi-Sensor Snow and Ice Mapping System (IMS) snow cover information to the snow assimilation system in 2004 (Zsótér et al., 2020). While the discontinuity around 2004 is present in the Eurasian watershed as well as in North America, the discontinuity around 1992 is only present in Eurasia. Figure 8 shows the spatial distribution of the snow melt discontinuities, calculated as differences between snow melt climatologies from 1979-1991 and 1992-2003, as well as 1992-2003 and 2004-2019. The latter difference exhibits spurious signals at the coastal mountain range of Alaska, the Rocky Mountains and at mountainous regions in Siberia, the prior discontinuity spots rather random signals only over Eurasia.

This discontinuity issue is not only limited to ERA5, but rather a general issue in reanalyses, as observation platforms are changing through time, making it practically very hard to make these products perfectly homogeneous in time. Especially satellite data were not available in the early days and were introduced with the development and introduction of new instruments. If the redundancy is large enough, then any discontinuity impact should be less pronounced. However, specifically for snow there is only the IMS product that was introduced in 2004, and hence any inhomogeneity generates a larger impact. Thus, this impact could possibly be reduced when using other data sets on top of the IMS product or instead of it, which ideally go further back in time.

## 4.3 Comparison with oceanic fluxes

In this section we look at the oceanic volume budget as described in Eq. (5). For this purpose we calculate volume fluxes through the oceanic gateways - BSO, Fram, Davis, Bering, Hecla and Fury Strait - using ocean reanalyses and compare them to our various freshwater input estimates. Oceanic storage change is derived from ocean reanalyses and from GRACE. To close the budget and get rid of any residuals we use a variational approach (see Sect. 2.1.2).
Figure 9 shows seasonal cycles of lateral net volume fluxes from ORAS5 and from the GREP ensemble for 1993-2018

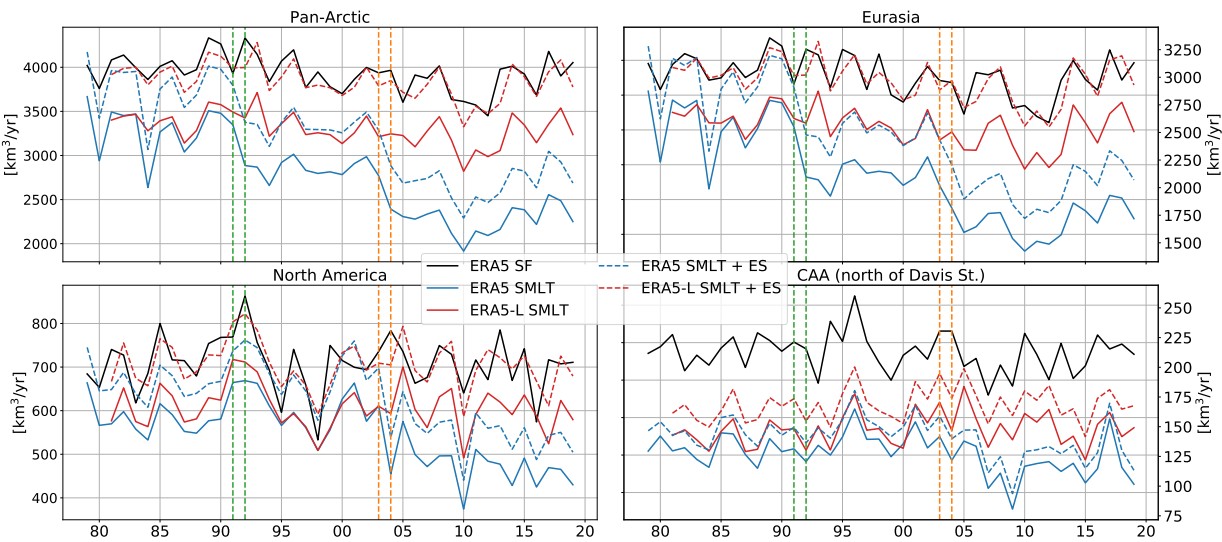

**Figure 7.** Annual means of ERA5 snowfall (SF, black; 1979-2019), ERA5 snow melt (SMLT, blue; 1979-2019) and the sum of ERA5 snow melt and snow evaporation (SMLT+ES, blue dashed; 1979-2019) as well as ERA5-Land snow melt (red; 1981-2019) and ERA5-Land SMLT+ES (red, dashed; 1981-2019) for the whole Pan-Arctic region (top left), Eurasia (top right), North America (bottom left) and the CAA (bottom right). Vertical lines show the position of the snowmelt discontinuities in 1992 (green) and 2004 (orange).

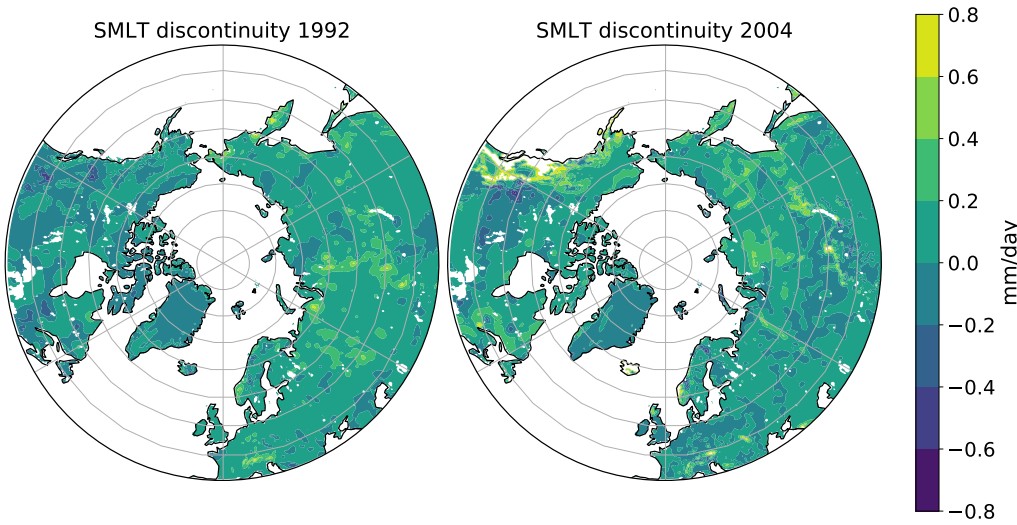

**Figure 8.** ERA5 snow melt discontinuities, calculated by subtracting the long term SMLT means from 1979-1991 and 1992-2003 and respectively 1992-2003 and 2004-2019.

The GREP ensemble shows an annual mean lateral transport out of the Arctic region of $203 \pm 48$ mSv, with ORAS5 being on the higher end of the ensemble with an annual outflow of $232 \pm 48$ mSv and a summer peak of 578 mSv. The seasonal cycles of the oceanic volume transports resemble the seasonal cycles of the freshwater input and peak in June, as the ocean reacts almost instantaneously to surface freshwater input and generates barotropic waves that lead to mass-adjustment within about a week (Bacon et al., 2015).

In addition to the oceanic transports the left panel of Fig. 9 shows the forcing terms involved in the generation of the ORAS5 fluxes:

$$F_{ORAS5} = (P_O - ET_O)_{ERAInterim} + R_{BT06} - \frac{\partial S_{ORAS5}}{\partial t} + damp_{ORAS5} + FW_{adj} \tag{11}$$

In addition to net precipitation from ERA-Interim and river discharge from the BT06 climatology, we need to add a surface salinity damping term ($damp_{ORAS5}$), which represents an additional non-physical surface freshwater forcing in the ocean re-analyses (last term of Eq. 6.1 in Madec et al., 2019). Additionally, a freshwater adjustment term ($FW_{adj}$) should be added due to the assimilation of global mean-sea-level changes. This term is not archived in ORAS5 and hence is not considered in our analysis (therefore it is marked grey in Eq. (11)). However, we deem the freshwater adjustment term as a rather small source of error and nevertheless would expect good closure concerning Eq. (11). Indeed, with an annual deviation of 11 mSv and an NRMSE value of 0.19 the accordance between $F_{ORAS5}$ and its freshwater input estimate is reasonable. In addition to the FW adjustment term, another cause for the discrepancies between forcing and computed volume fluxes is that the ocean reanalyses compute their own turbulent air-sea fluxes (REF) and do not use that from ERA-Interim. However, given the generally low values of sensible and latent heat fluxes in the Arctic we consider this a moderate source of error.

The right panel of Fig. 9 shows various estimates of freshwater input minus oceanic storage change. To estimate the atmospheric freshwater input over the ocean we take net precipitation from ERA5 as well as VIWVD from ERA5, ERA-Interim, the Japanese 55-year Reanalysis JRA55 and JRA55-C. For the reanalysis-based estimates we use the river discharge reanalyses we have most confidence in, namely GloFAS$_{E5_{new}}$, ERA5-Land runoff and the indirect estimates through P-ET and VIWVD minus storage change. Land storage is only derived from GRACE, while oceanic storage is taken from GRACE and from ORAS5. GRACE and ORAS5 show quite similar seasonal cycles, with mass being accumulated by the ocean in summer and released in winter, most likely caused by the seasonal variation of wind stress curl and seasonal changes in Ekman pumping (Bacon et al., 2015). In terms of annual trends both ORAS5 and GRACE indicate a slight increase of mass over the past decades. The observation-based estimates are calculated using river discharge from our observatiom-estimates (*ae, Ee, Ge*) and oceanic storage change from GRACE. The atmospheric components VIWVD and atmospheric storage change over the ocean are still taken from reanalyses (ERA5, ERA-Interim, JRA55, JRA55c), as we lack the corresponding observations. Greenlandic discharge is taken from Mankoff et al. (2020a, b). Calculating the mean over our reanalysis- and observation-based estimates we obtain annual values of $207 \pm 4$ mSv and $208 \pm 3$ mSv, respectively. Thus, compared to the GREP volume flux we obtain an imbalance of $4 - 5$ mSv, representing less than 3% of the physical fluxes.

Figure 10 shows the budget residuals, calculated as GREP ensemble average volume transport minus the various freshwater estimates. From January to April residuals are generally positive, as the freshwater input from atmosphere and land is low due

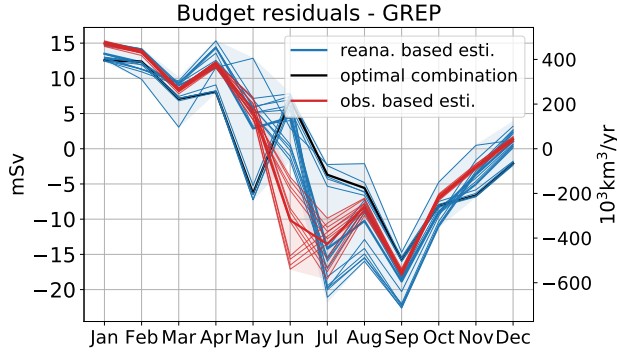

**Figure 9.** Left panel: Mean seasonal cycles of lateral net volume fluxes (defined positive out of the Arctic) from ORAS5 (1993-2018) and GREP (1993-2018), as well as the sum of forcing terms generating ORAS5. Right panel: Different realisations of observation- and reanalysis-based estimates of freshwater input minus oceanic storage for the period 1993-2018.

**Figure 10.** Mean annual cycles of different realisations of the budget residuals (all 1993-2018; reanalyses based: blue lines, observation based: red lines) and the optimal combination yielding the smallest NRMSE value (black line).

| | GREP | ORAS5 | ORAS5 forcing | ARCGATE | reana. based estimate | obs. based estimate |
|---|---|---|---|---|---|---|
| $\mu$ [mSv] | 203/200* | 232/221* | 243 | 151* | 207 | 208 |
| NRMSE | - | $0.22^1$ | $0.34^1/0.19^2$ | $0.79^{1*}/0.83^{2*}$ | $0.50^1/0.38^2$ | $0.52^1/0.35^2$ |
| r | - | $0.97^1$ | $0.87^1/0.94^2$ | $-0.39^{1*}/-0.52^{2*}$ | $0.64^1/0.77^2$ | $0.74^1/0.87^2$ |

**Table 8.** 1993-2018 ($^*$2004-2010) long term means, NRMSEs and correlation coefficients of oceanic volume transports from GREP, ORAS5 and ARCGATE, as well as estimates derived from the forcing terms used in ORAS5 (as described in Eq. (11)). Additionally, our reanalyses and observation based estimates are given. NRMSE values and correlation coefficients are calculated using monthly means with respect to GREP($^1$) and ORAS5($^2$). Units are milli-Sverdrup (mSv).

to river icing and low precipitation values, however freshwater that entered the Arctic in the past season and moves slowly with the ocean currents still finds its way through the Arctic gateways. In June, the large freshwater peak generates barotropic waves that are also seen in the reanalysis volume transports. Residuals arise from different river discharge estimates and are
generally positive for reanalysis- and negative for observation-based estimates. In late summer, river discharge tends to decline and precipitation takes over the main role of delivering freshwater to the ocean. Oceanic reanlyses show a fast decline after the June peak and exhibit a volume transport minimum in September leading to a negative residual peak in all estimates. We assume that volume that enters the Arctic as freshwater through precipitation is slowly transported with the ocean currents and modified by wind stress, partly freezes on the way and takes weeks up to months to leave the Arctic area, explaining the
elevated winter and spring transports.

Table 8 shows annual means of the various transport estimates, standard deviations and NRMSE values. The normalised RMSE values are calculated with respect to GREP ($^1$) and ORAS5 ($^2$), indicated by the superscripts. In addition to the estimates from Fig. 9, also the observation-based mean volume transport from the ARCGATE project (Tsubouchi et al., 2019; Tsubouchi et al., 2012, 2018) is given. To compare the ARCGATE value to ORAS5 and GREP, mean values and NRMSEs are calculated
over the ARCGATE period (10/2004 - 05/2010, indicated by a $^*$). With an annual mean volume transport of 151 mSv the ARCGATE estimate is about 25% lower than the GREP estimate. The high NRMSE values emerge as the observation based ARCGATE flux does not show any peaks in June but rather stays low throughout the summer (not shown). This is probably a result from the mooring arrays in the gateways being too sparse and hence the velocity field not being measured accurately enough to dissolve the barotropic wave signal. Instead ARCGATE shows the export of freshwater that travels with the oceanic
currents, dilutes on the way and remains around the continental shelfs for some time. Hence, the oceanic reanalyses and the observation based ARCGATE estimate should only be compared in terms of annual means and not seasonalities.

### 4.3.1 Volumetric budget closure

To close the volumetric budget and get rid of the small residual we use a variational adjustment procedure (see Sect. 2.1.2).
A priori estimates are calculated by taking the mean over all trustworthy estimates of the individual budget terms - given in

Table 9. We perform the adjustment on long term annual means to close the annual budget and on the monthly climatologies of the individual terms to close the budget on a monthly scale. The uncertainties for the annual optimization are estimated as the standard deviations between the various estimates (see Table 9) of the mean annual budget terms. The uncertainties for the monthly optimization are estimated by taking the annual maximum of the monthly standard deviations, indicated by the dashed lines in Fig. 11. Oceanic lateral volume transport F features the largest uncertainties and hence also experiences the largest adjustments amongst all budget terms.

Figure 12 shows the adjusted long term annual volume budget over the oceanic and land domains. Oceanic storage indicates

| | |
|---|---|
| F | GREP |
| R | $Obs_{Ge}$, $GloFAS_{E5_{new}}$ ,ERA5-Land |
| $R_{Greenland}$ | Liquid and solid from Mankoff et al. (2020a, b) |
| $Ocean_{Atm,in}$ | VIWVD from ERA5, ERA-Interim, JRA55, JRA55c |
| $Ocean_{stor}$ | GRACE (CSR,GSFC,JPL) |
| $Land_{Atm,in}$ | VIWVD from ERA5, ERA-Interim, JRA55, JRA55c |
| $Land_{stor}$ | GRACE (CSR,GSFC,JPL) |

**Table 9.** List of employed datasets in the variational adjustment procedure.

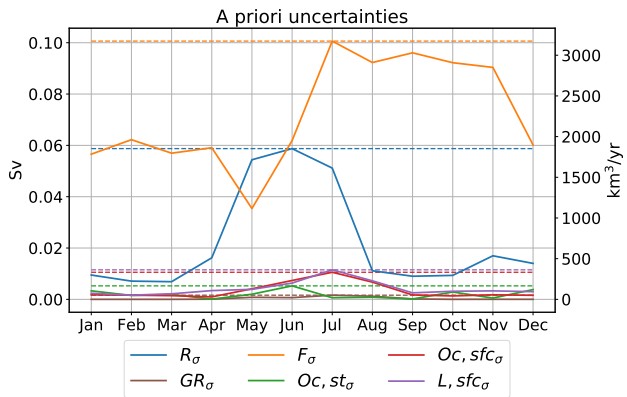

**Figure 11.** Mean annual cycles (1993-2018) of spread for runoff (R), Greenland runoff (GR), lateral ocean fluxes (F), oceanic storage (Oc,st) and atmospheric input over ocean (Oc,sfc) and land (L,sfc). Spread is calculated as standard deviate between various estimates (see table 9) of the individual terms and as RMS error estimate for GRACE.

a slight increase ($28 \pm 6 km^3$) of mass over the past decades. About $6532 \pm 84$ km$^3$ leave the Arctic Ocean through the main gateways on an annual basis. From the water entering the Arctic Ocean through its surface, roughly two thirds are supplied by runoff ($4379 \pm 25$ km$^3$) and roughly one third ($2181 \pm 80$ km$^3$) is freshwater delivered by the atmosphere. Oceanic volume transports out of the Arctic domain exceed the atmospheric moisture entering the Arctic ($6314 \pm 121$ km$^3$) by nearly 4%, indicating an annual loss of water volume of roughly 220 km$^3$. This loss is mainly driven by terrestrial water mass losses. Even though the representation of frozen land components is not ideal in HTESSEL, the comparison of GRACE mass changes to the

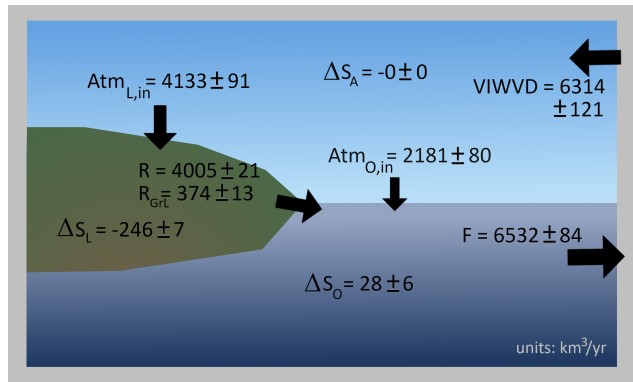

**Figure 12.** 1993-2018 adjusted long term means of atmospheric and terrestrial freshwater fluxes and storage terms in the Arctic hydrological cycle, as well as oceanic volume transport and storage change. Units are km$^3$ per year; arrow areas are scaled by the magnitude of the represented terms.

sum of ERA5 storage changes (snow and soil water) and glacial changes taken from literature (e.g., Wouters et al., 2019) agree well. Therefore, we combine land storage changes from ERA5 (excluding glaciers) with storage changes from GRACE to estimate contributions of different terrestrial sources to the diagnosed storage decline in the Arctic. We find that approximately 47% of the 246 km$^3$yr$^{-1}$ decline are generated through liquid and solid discharge from Greenland, while about 45% come from Arctic glaciers (excluding Greenland) and the remaining eight percent are the result of a decline in land water storage due to permafrost and snow cover reduction.

The reported increase in oceanic storage is driven by mass increases over the western Arctic Ocean and the coastal areas of Eurasia and North America (see Fig. A1 in the appendix). For the western Arctic Ocean, various studies indicate an accumulation of freshwater in the Beaufort Gyre, due to a combination of favourable wind forcing, redirection of Mackenzie River discharge, inflow of low salinity waters through Bering Strait and sea ice melt (e.g., Proshutinsky et al., 2019). Mass increases along the coastal areas are a result of runoff increases (Ludwigsen et al., 2020; Morison et al., 2012). Mass decreases in Barents and Kara Sea , as well as in Baffin Bay (Fig. A1 in the appendix) counteract those increases, resulting in the reported weak trends for the whole Arctic Ocean. Mass decreases west of Greenland are mainly caused by lowering of the geoid associated with nearby ice mass losses. However, Jeon et al. (2021) also found residual land-leakage effects that were not removed in the GRACE Mascons. Furthermore, oceanic storage is strongly affected by decadal wind variations (Volkov and Landerer, 2013; Fukumori et al., 2015) and circulation patterns and exhibits strong nonseasonal fluctuations, further complicating the detection of long-term oceanic mass trends. Volkov and Landerer (2013) find that an intensification of the westerly winds over the North Atlantic and over the Russian Arctic continental shelf lead to a decrease of ocean mass in the central Arctic. Further, they found positive correlation between Arctic Ocean mass fluctuations and northward wind anomalies over the Bering Seas and the northeastern North Atlantic. They also reveal that cyclonic/anticyclonic anomalies of the large-scale ocean circulation lead to negative/positive Arctic ocean mass anomalies.

Table 10 provides adjusted monthly and annual climatologies and Fig. 13 shows seasonal cycles of the oceanic budget terms

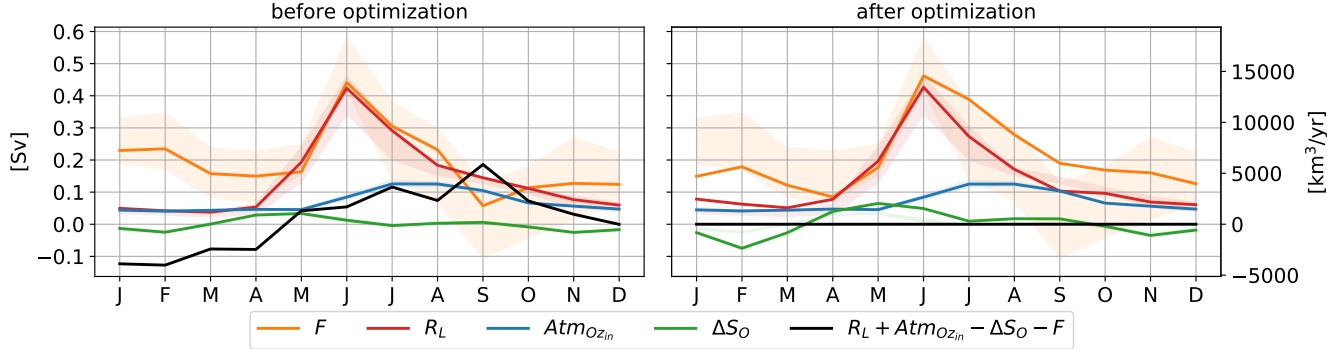

**Figure 13.** 1993-2018 adjusted mean annual cycles of atmospheric and terrestrial freshwater fluxes in the Arctic hydrological cycle as well as oceanic volume transports and storage changes before (left) and after (right) the optimization. Shading represents the uncertainties of the a priori estimates (shown in both plots). The black line shows the sum of all budget terms and by going to zero it demonstrates the closure of the monthly budgets after the optimization.

before and after the adjustments. A metric to identify whether the variational adjustment is successful is the comparison of the adjustments of the terms to their respective a priori uncertainty (L'Ecuyer et al., 2015), hence Fig. 13 also shows the spreads of the a priori estimates (shaded areas). Due to the large differences between oceanic volume transports and freshwater input terms in late winter to early spring and in September, both F and runoff R feature adjustments larger than their a priori spreads,

suggesting that the a priori uncertainties are larger than estimated. A possible explanation is that systematic biases are not taken into account. The state-of-the art reanalyses exhibit systematic errors in their runoff seasonalities, as the seasonal runoff peaks in summer are too low in comparison to observations, while winter and spring values are too high. Due to the lack of reliable seasonal observations of the oceanic volume fluxes, it is hard to define systematic biases in the ocean reanalyses. However, all four ocean reanalyses feature quite low September volume fluxes, which are not found in their forcing components (see

figure 9). Uotila et al. (2019) assess ten ocean reanalyses, including CGLORS, FOAM, GLORYS and ORAS5, specifically in the polar regions and find multiple systematic errors concerning sea ice thickness and extent, temperature profiles, mixed layers as well as ocean transports. Seasonal cycles of volume transports were not assessed, however seasonal cycles of sea ice components and heat transports did exhibit systematic errors. Further analyses would be necessary to come up with robust estimates of the bias in seasonal volume transports.

|  | R | GR | $\text{Atm}_{O,in}$ | $\Delta\text{S}_O$ | F |
|---|---|---|---|---|---|
| Jan | 2264 | 204 | 1415 | -820 | 4703 |
| Feb | 1770 | 203 | 1305 | -2361 | 5640 |
| Mar | 1412 | 203 | 1385 | -830 | 3830 |
| Apr | 2240 | 207 | 1487 | 1252 | 2682 |
| May | 5932 | 264 | 1433 | 2046 | 5583 |
| Jun | 12775 | 679 | 2659 | 1543 | 14570 |
| Jul | 7376 | 1242 | 3943 | 301 | 12260 |
| Aug | 4764 | 628 | 3943 | 543 | 8792 |
| Sep | 3019 | 234 | 3265 | 531 | 5988 |
| Oct | 2826 | 210 | 2075 | -203 | 5313 |
| Nov | 1973 | 207 | 1772 | -1098 | 5050 |
| Dec | 1708 | 205 | 1491 | -572 | 3976 |
| Mean | 4005 | 374 | 2181 | 28 | 6532 |

**Table 10.** 1993-2018 adjusted mean annual cycle of the Arctic Oceans hydrological cycle; units are km$^3$ per year.

## 5 Conclusions

We analysed and compared various estimates of runoff into the Arctic Ocean on a seasonal and annual basis and found considerable differences in terms of seasonalities, mean values and trends. Further, we used a non-steric formulation of the Arctic's Ocean volume budget equation and compare freshwater input into the ocean to lateral volume transports over the Arctic's boundaries. To close the budget and get best estimates of all budget terms we applied a variational adjustment procedure. The main outcomes of this study are the following:

– River discharge observations for the major Arctic catchments show distinct runoff peaks in June due to snow melt and river ice breakup and low runoff values in winter. Annual trends indicate slight discharge increases over the past decades - the largest increases evident at the Lena and Ob basins with a rise of about $4\pm2\%$ per decade. The total Pan-Arctic area (excluding Greenland) exhibits an upward trend of $2\pm1\%$ per decade. Holmes et al. (2018) show that the six largest Eurasian rivers (Ob, Yenisei,Lena, Kolyma, Pechora, Severnaya Dvina) exhibit an increase of $3.3\pm1.6\%$ and the two largest North American rivers (Mackenzie and Yukon) an increase of $2.0\pm1.8\%$ per decade.

– We estimate Pan-Arctic river discharge from gauge observations using monthly correction factors from $\text{GloFAS}_{E5_{new}}$, as the common method of hydrological analogy (e.g., Shiklomanov and Shiklomanov, 2003) tends to underestimate the high flow summer peaks (see Fig. 4), and obtain a long term annual flux of $4031\pm203$ km$^3$ (excluding Greenland). To compare our results to past studies we adapted the time periods and areal extents accordingly and found reasonable accordance with Haine et al. (2015), who combined runoff from ERA-Interim with river discharge observations and obtained a total discharge about 5% higher than our estimate. An even better agreement was found with the estimates made by Shiklomanov et al. (2021a), as the total Pan-Arctic discharges (including Greenland) agree within 2%.

- Runoff from ERA5 is underestimated compared to observed discharge, with the largest discrepancies occurring in the high flow summer months, and features strong declines of $10\pm1\%$ per decade on a Pan-Arctic basis and even stronger declines at the major catchments. These strong declines are caused by two inhomogeneities (1992 and 2004) in ERA5's snow melt time-series and contradict the discharge increases found in gauge observations. Those inhomogeneities are caused by a loss of snow through changes in the data assimilation system. While the discontinuity around 2004 was traced back to the introduction of the IMS snow cover information to the snow assimilation system, the 1992 inhomogeneity still needs further investigation.

  ERA5-Land does not feature these inhomogeneities and exhibits more moderate runoff declines of $5\text{-}6\pm2\%$ for the Eurasian major watersheds and $2\pm1\%$ for Mackenzie river and on the Pan-Arctic scale. These declines in ERA5-L runoff are caused by similar declines in P-E from ERA5, as P-E is used as a forcing in ERA5-L (see figure 6 and table 5). As observations agree on an increase of river discharge, these declines are deemed unrealistic. An improvement may possibly be achieved when taking the divergence of moisture flux (VIWVD) as forcing, as VIWVD, which is computed from analysed fields rather than short-term forecasts, features similar trends as discharge observations.

- Calculating runoff through the divergence of moisture flux is the only reanalysis-based estimate that exhibits a slightly positive trend of $2\pm1\%$ per decade and thus features the best agreement with observations in terms of trends and the highest correlation coefficient. However, VIWVD tends to underestimate runoff by roughly 5%.

- River discharge from $\text{GloFAS}_{E5}$ reflects the variation and the inhomogeneities from ERA5 and shows an additional negative shift due to two sink terms removing water in LISFLOOD that lead to a discharge underestimation of up to 50% towards the end of the time series. In contrast, $\text{GloFAS}_{E5_{new}}$ entails a considerable improvement and reproduces the observed values best in terms of annual means and NRMSE values.

- Liquid and solid discharge from Greenland account for roughly 10% of total Pan-Arctic discharge and hence should not be neglected in assessments of the Arctic freshwater budget. Liquid discharge features a vast increase of $20\pm4\%$ per decade and solid discharge an increase of $8\pm1\%$. Due to glacial melting also land storage changes are particularly strong over the Greenlandic ice sheet, accounting for roughly half of total Pan-Arctic land storage change.

- Comparing the estimates of freshwater input into the Arctic Ocean that we have most confidence in after the preceding analysis (listed in table 7), to oceanic volume transports through the Arctic gateways computed from ocean reanalysis yields only a small imbalance of less than 3% in terms of annual means.

- The variational adjustment procedure provides a best estimate of every budget term for every calendar month - listed in Tab. 10. About two thirds of the freshwater provided for oceanic volume transports come from runoff ($4379\pm25$ km$^3$ per year) and about one third is provided by the atmosphere. Land areas show a strong decline ($-246$ km$^3$yr$^{-1}$) of water storage over the past decades, while oceanic storage features only a slight increase. This leads to a surplus of roughly 220 km$^3$ per year of more water leaving the Arctic area than entering.

In summary, we refined past Arctic water budget estimates (e.g., Serreze et al., 2006; Dickson et al., 2007) and their uncertainties by combining some of the most recent reanalyses data-sets and observations, and by applying a variational optimization scheme. The variational adjustment worked very well on an annual scale and brought reliable estimates of the volume budget terms, requiring only moderate adjustments of less than 3% for each individual term. Adjustments are considered reliable if budget closure is achievable within the respective terms error bounds and if the terms are comparable to estimates from past studies. With an annual value of $4379 \pm 25$ km$^3$ (calculated over 1993-2018), our adjusted runoff estimate is slightly higher than estimates made by Shiklomanov and Lammers (2013) and Shiklomanov et al. (2021a) for the period 1936-2006. However, considering the different calculation periods and assuming a decadal rise of roughly 2%, the estimates are considered to be in good agreement. On a seasonal scale however, stronger adjustments were needed to close the budget, and some of the adapted freshwater and volume fluxes fell out of their a priori uncertainty range, suggesting an underestimation of the specified uncertainties. The latter is very likely caused by the presence of systematic errors being present in the data sets, or at least in their seasonal cycles, that are not taken into account in our a priori uncertainty estimates. Especially when calculating Pan-Arctic runoff, caution is needed. Our results show that seasonal peaks of river discharge are underestimated in almost all of the assessed reanalyses (ERA5, ERA5-Land, GloFAS$_{E5}$, GloFAS$_{E5L}$). The biggest errors are caused by inhomogeneities in the data assimilation system (ERA5 and GloFAS$_{E5}$) which led to a great underestimation of runoff, especially in the latter half of the time series. However, also reanalyses without data assimilation (ERA5-Land and GloFAS$_{E5L}$) were not able to reproduce the seasonal cycle of river discharge accurately. On the other hand we find distinct improvements in the new GloFAS$_{E5_{new}}$ product, especially when investigating seasonal cycles and long term means it features considerable enhancements compared to its precursors. However, for interannual variability and trend analysis we recommend the use of the VIWVD estimate, as it reproduces trends from gauge observations more accurately than the other estimates.

When extrapolating observed river discharge to the whole Pan-Arctic area we found that the common method of hydrological analogy tends to underestimate the discharge peaks. We therefore advise to use river discharge observations where available and reliable runoff/discharge estimates from reanalyses (e.g., GloFAS$_{E5_{new}}$ or ERA5-Land) to extrapolate discharge to the ungauged areas.

A further possible reason for inconsistencies between runoff and ocean reanalyses is the usage of climatological river discharge data to specify land freshwater input in ocean reanalyses. Our analyses show that the seasonal cycle of the ORAS5 runoff climatology BT06 fits well to our observation based estimates (see Fig. 5), however the lack of inter-annual variability in the freshwater input adversely affects i.a. oceanic volume transports. We note that Zuo et al. (in preperation) work on implementing a time-varying land freshwater input, derived from discharge data from GloFAS, into ORAS5. This should further reduce the inconsistencies between runoff and oceanic volume fluxes from ocean reanalyses.

To further refine the budget estimates, longer time series of all budget terms would be needed. For example, one could repeat the analysis using the back extension of ERA5 which goes back to 1950. There is also a new bias-corrected ERA5 data set (WFDE5, Cucchi et al., 2020), that could be examined in terms of the Arctic water budget. Further, it would help to include a precipitation observation data set, preferably one that combines available satellite-based and gauge-based data sets.

Concerning estimation of biases in ocean reanalyses, one could in principle draw on information from oceanographic data

for comparison. A main difficulty with oceanographic data is the generally limited temporal and spatial coverage. Nevertheless, the unique form of the Arctic Ocean (as water leaves and enters only through a handful of gateways) allows relatively easy measurements of the in- and outgoing fluxes. As an example, measurements from arrays of moored instruments (like e.g. Acoustic Doppler Current Profilers, MicroCAT – CTD Sensors and Seagliders) were taken to estimate transports through
the Arctic gateways using a mass-consistent framework (Tsubouchi et al., 2012, 2018). Our results however showed that the moored instruments did not measure the velocity field accurately enough to resolve the barotropic wave signal arising from temporally varying runoff (Tsubouchi, 2019) leading to errors in the seasonality of the net volume flux. A longer measuring period with an even denser monitoring network could help with this aspect.

*Code and data availability.* ERA5 and ERA5-Land data are available on the Copernicus Climate Change Service (C3S) Climate Data Store
(Hersbach et al., 2019; Muñoz Sabater, 2019). GloFAS river discharge data are also available on the Copernicus Climate Change Service (C3S) Climate Data Store (Harrigan et al., 2019). GRACE monthly ocean bottom pressure anomalies and land water-equivalent-thickness anomalies are available through the Physical Oceanography Distributed Active Archive Center (Landerer, 2020b, a). River gauge observations are downloadable through the Arctic Great Rivers Observatory (Shiklomanov et al., 2021b) and the Regional Arctic Hydrographic Network data set (Lammers et al., 2001).

**Appendix A: Additional figures**

Fig. A1 shows trends of terrestrial and oceanic water storage changes derived from GRACE Mascons.

**Appendix B: List of Acronyms**

**ARCGATE** Mooring-derived data of oceanic fluxes through the Arctic gateways

**BT06** runoff climatology used in ORAS5

**C3S** Copernicus Climate Change Service

**CAA** Canadian Arctic Archipelago

**CDS** Climate Data Store

**CGLORS** ocean reanalyses from the Euro-Mediterranean Center on Climate Change

**CMEMS** Copernicus Marine Environment Monitoring Service

**CEMS** Copernicus Emergency Management Service

**ECMWF** European Centre for Medium-Range Weather Forecasts

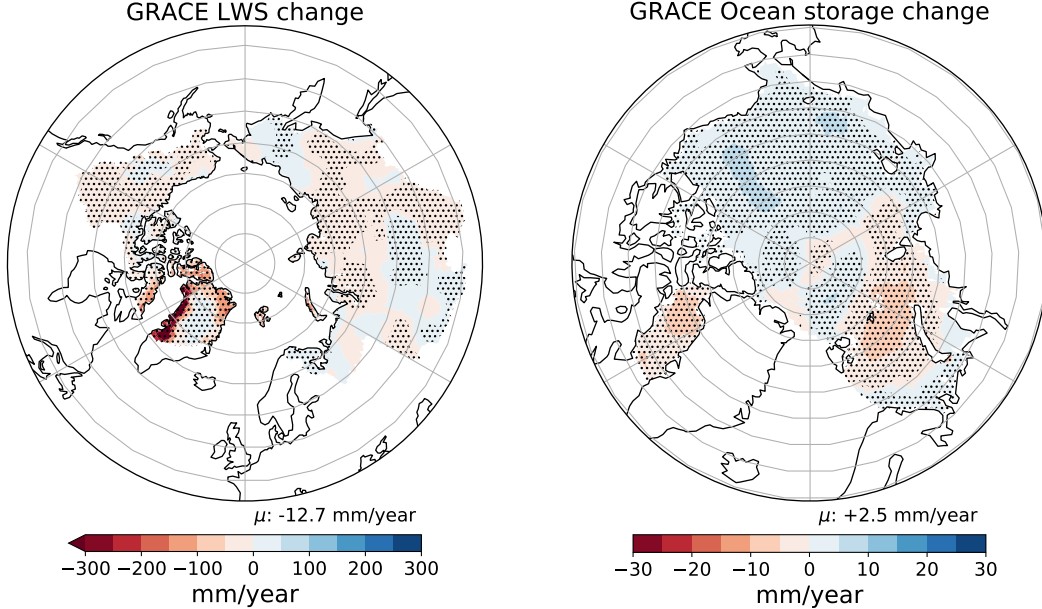

**Figure A1.** 2002-2019 land water storage trends (left, areal mean of -12.7 mm/year) and oceanic storage trends (right, areal mean of +2.5 mm/year), calculated as mean of GRACE JPL, CSR and GSFC Mascons. Shading denotes significant trends (p values < 0.05).

**ERA5** ECMWF's fifth atmospheric reanalysis

**ERA5-Land** offline simulation of ERA5

**ERA-Interim** ECMWF interim reanalysis

**FOAM** ocean reanalyses from the UK Met Office

**GloFAS** Global Flood Awareness System

**GLORYS** ocean reanalyses from Mercator Ocean

**GRACE** Gravity Recovery and Climate Experiment

**GREP** Global ocean Reanalysis Ensemble Product

**HTESSEL** Hydrology Tiled ECMWF Scheme for Surface Exchanges over Land

**JRA55** Japanese 55-year Reanalysis

**JRC** Joint Research Centre

**LISFLOOD** model for river basin scale water balance and flood simulation

**NEMO**  Nucleus for European Modelling of the Ocean

**NRMSE**  normalized root mean square error

**ORAS5**  ECMWF's Ocean Reanalysis System 5

**r**  Pearson's correlation coefficient

$c$  horizontal sea water velocity

**ET**  evapotranspiration

**F**  oceanic lateral volume transport

$\mathbf{F}^{surf}$  surface freshwater fluxes (precipitation, evaporation, runoff)

$g$  gravitational constant

**P**  precipitation

$p_s$  surface pressure

$q$  specific humidity

**R**  runoff or river discharge

$rho_w$  density of freshwater

$\mathbf{S}_A$  atmospheric storage

$\mathbf{S}_L$  land storage

$\mathbf{S}_O$  ocean storage

**SD**  snow depth

**SWVL**  Volumetric soil water per layer

$v$  horizontal wind vector

**VIWVD**  Vertical integral of divergence of moisture flux

*Author contributions.* S.W. performed the data analysis, including the production of the figures in the paper and prepared the manuscript. M.M., V.S., E.Z., H.Z. and L.H. contributed to the interpretation of results and the writing of the manuscript.

*Competing interests.* The authors declare that they have no conflict of interest.

*Acknowledgements.* The authors thank Takamasa Tsubouchi for the helpful discussions on ARCGATE data. S.W., V.S., and M.M. were
supported by Austrian Science Fund project P33177. V.S. was additionally supported by Copernicus Marine environment Service contract
114- R&D-GLO-RAN-CMEMS Lot 8.

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
