# Peer review of "Diagnostic evaluation of river discharge into the Arctic Ocean and its impact on oceanic volume transports"

_Hydrology and Earth System Sciences, 2021_

## Author Comment (AC1)

**Response to Anonymous Referee #1**

Thank you very much for your positive comments and constructive feedback, you addressed some important points. Your clarifications helped to make the manuscript clearer for the reader. Our responses are provided in blue together with your original comments in black.
We really appreciate your time and insight in reviewing our manuscript!

Kind regards,
Susanna (on behalf of all co-authors)

General/overarching comments:

*To answer your general comments, we revised the whole conclusion section (from line 555 onward) and marked adapted passages* **bold**:

**Summarizing we refined past Arctic water budget estimates (Serreze et al., 2006; Dickson et al., 2007, e.g.,) and their uncertainties by combining some of the most recent reanalyses datasets and observations, and by applying a variational optimization scheme. The variational adjustment worked very well on an annual scale and brought reliable estimates of the volume budget terms, requiring only moderate adjustments of less than 3% for each individual term. Adjustments are considered reliable if budget closure was achievable within the respective terms error bounds and if the terms were comparable to estimates from past studies.** With an annual value of 4379 ± 25 km3 (calculated over 1993-2018), our adjusted runoff estimate is slightly higher than estimates made by Shiklomanov and Lammers (2013) and Shiklomanov et al. (2021a) for the period 1936-2006. However, considering the different calculation periods and assuming a decadal rise of roughly 2%, the estimates come quite close. On a seasonal scale however, the adjustment process was not a full success, as the budget residuals of some months are too large to be eliminated within the a priori spreads of the individual terms. This is very likely caused by systematic errors being present in the datasets, or at least in their seasonal cycles, that are not considered in our a priori uncertainty estimates.
**Especially when calculating Pan-Arctic runoff, caution is needed. Our results show that seasonal peaks of river discharge are underestimated in almost all of the assessed reanalyses (ERA5, ERA5-Land, GloFASE5, GloFASE5L). The biggest anomalies are caused due to inhomogeneities in the data assimilation system (ERA5 and GloFASE5) which led to a great underestimation of runoff, especially at the latter half of the time series. However also reanalyses without data assimilation (ERA5-Land**

**andGloFASE5L) were not able to reproduce the seasonal cycle of river discharge accurately.** On the other hand, we find distinct improvements in the new GloFASERA5newproduct, especially when investigating seasonal cycles and long term means it features vast enhancements compared to its precursors. However, for interannual variability and trend analysis we would recommend the use of the VIWVD estimate, as it reproduces trends from gauge observations quite accurately. **When extrapolating observed river discharge to the whole Pan-Arctic area we found that the common method of hydrological analogy tends to underestimate the discharge peaks. We therefore advise to use river discharge observations where available and reliable runoff/discharge estimates from reanalyses (e.g., GloFASERA5new or ERA5-Land) to extrapolate discharge to the ungauged areas.**

A further possible reason for inconsistencies between runoff and ocean reanalyses is the usage of climatological river discharge data to specify land freshwater input in ocean reanalyses. Our analyses show that the seasonal cycle of the ORAS5 runoff climatology Bt06 fits quite well to our observation-based estimates (see Fig. 5), however the lack of inter-annual variability in the freshwater input alters i.a. oceanic volume transports. In their BRONCO (Benefits of dynamically modelled river discharge input for ocean and coupled systems) project Zuo et al. (in preparation) work on implementing a time-varying land freshwater input, derived from discharge data from GloFAS version 2.1, into ORAS5. This should further reduce the inconsistencies be-tween runoff and oceanic fluxes from ocean reanalyses.

**To further refine the budget estimates, longer time scales of all budget terms would be needed. For example, one could repeat the analysis using the back extension of ERA5 which goes back to 1950. There's also a new bias corrected ERA5 data set (Cucchi et al., 2020, WFDE5), that could be examined in relation to the Arctic water budget. Further it would help to include a precipitation observation data set, preferably one that combines available satellite-based and gauge-based data sets. Concerning the biases in ocean reanalyses, one could refer to oceanographic data for comparison. Generally, comparison to oceanographic data is difficult, as observations are limited concerning their temporal and spatial coverage. Nevertheless, the unique form of the Arctic Ocean (as water leaves and enters only through a handful of gateways) allows relatively easy measurements of the in- and outgoing fluxes. This was done in the ARCGATE project, where data from arrays of moored instruments (like e.g., Acoustic Doppler Current Profilers, MicroCAT – CTD Sensors and Seagliders) were taken to estimate transports through the Arctic gateways. Our results showed that it is possible to estimate annual fluxes into and out of the Arctic Ocean quite accurately, however the moored instruments did not measure the**

**velocity field accurately enough to resolve the barotropic wave signal arising from temporally varying runoff (T. Tsubouchi, personal communication 2021) leading to differences in the seasonality of the fluxes. A longer measuring period with an even denser monitoring network could help with this aspect.**

1. Although river discharge data and land/ocean water storage data are used carefully, the paper doesn't use oceanographic data to estimate the marine water fluxes (it only uses reanalysis data). This isn't a major problem because the paper focuses on runoff and river discharge, but it should be mentioned and discussed somewhere (Conclusions?).

   We discussed it at the end of the revised conclusion above.

2. Related to point 1: What's the scope/opportunity for future improvements on the Arctic water budget analysis?  What model and data assimilation improvements would help?  What data are needed to refine the budget estimates?  Again, this isn't a major omission, but it will help set the context for future work if this point is discussed somewhere (Conclusions?).

   What's the scope/opportunity for future improvements on the Arctic water budget analysis?
   See revised conclusion above.

   What model and data assimilation improvements would help?
   *We added the following paragraph in section 4.2.3:*
   "This discontinuity issue is not only limited to ERA5, but rather a general issue in reanalyses, as observation platforms are changing through time, making it practically very hard to make these products 100% homogeneous in time. Especially satellite data were not available in the early days and were introduced with the development and introduction of new instruments. If the redundancy is large enough, then any discontinuity impact should be less pronounced. However, specifically for snow there's only the IMS product that was introduced in 2004, and hence any inhomogeneity generates a larger impact. Thus, this impact could be reduced when using other data sets on top of the IMS product or instead of it, which ideally go further back in time."

   What data are needed to refine the budget estimates?
   See revised conclusion above. The part about longer time scales, the bias corrected ERA5 and precipitation observations.

Specific Comments:

1. Line 50: Cite where it says the ERA5 runoff features spurious trends.

We added a citation of Zsoter et al. (2020)

2. Figures 1 and 2: What is the source of catchment data in Figures 1 and 2?

   We added the following catchment sources (shapefiles):

   **Individual river catchment outlines** were taken from the CEO Water Mandate Interactive Database of the World's River Basins (http://riverbasins.wateractionhub.org/)
   **Regional outlines** (CAA etc.): GRDC (2020): WMO Basins and Sub-Basins / Global Runoff Data Centre, GRDC. 3rd, rev. ext. ed. Koblenz, Germany: Federal Institute of Hydrology (BfG).

3. Section 2: Add a table containing information on the runoff and discharge sources (ERA5, ERA5-Land, GloFAS..., GREP etc.)

   We added a table containing all runoff and river discharge sources and key information of the individual products.

| Product | Description | Variable | Period | |
|---|---|---|---|---|
| ERA5 | Fifth generation ECMWF reanalysis using IFS (+ HTESSEL) | Runoff [m/s] | 1979-2019 (back extension to 1950 available) | Hersbach et al., 2020 |
| ERA5-Land | Offline simulation of ERA5 without DA using HTESSEL | Runoff [m/s] | 1981-2019 (back extension to 1950 expected in 2021) | Muñoz-Sabater et al., 2021 |
| GloFAS$_{E5}$ | ERA5 runoff + simplified LISFLOOD | River discharge [m$^3$/s] | 1979-2019 | Harrigan et al., 2019 |
| GloFAS$_{E5L}$ | ERA5-Land runoff + simplified LISFLOOD | River discharge [m$^3$/s] | 1999-2018 | - |
| GloFAS$_{E5new}$ | Full configuration of LISFLOOD | River discharge [m$^3$/s] | 1979-2019 | - |
| Bt06 | Runoff climatology used in ORCA025 | River discharge [m$^3$/s] | Climatology | Bourdalle-Badie and Treguier, 2006 |

| Observations | Measurements at gauging stations | River discharge [$m^3$/s] | - | - |
| --- | --- | --- | --- | --- |

New citation:
Muñoz-Sabater, J., Dutra, E., Agustí-Panareda, A., Albergel, C., Arduini, G., Balsamo, G., Boussetta, S., Choulga, M., Harrigan, S., Hersbach, H., Martens, B., Miralles, D. G., Piles, M., Rodríguez-Fernández, N. J., Zsoter, E., Buontempo, C., and Thépaut, J.-N.: ERA5-Land: a state-of-the-art global reanalysis dataset for land applications, Earth Syst. Sci. Data, 13, 4349–4383, https://doi.org/10.5194/essd-13-4349-2021, 2021.

4. Line 70: "river discharge" includes both liquid water and ice (presumably)?

   Yes, it does, we clarified it.

5. Line 73: For clarity, say that "associated domain" means the catchment area.

   We clarified that.

6. Line 104: It talks about "different bulk formulas and differences in the data assimilation…" Different to what? Be specific.

   *We specified some of the differences and referred to the individual documentations for further details. The following paragraph was added after line 103:*

   "While the GREP ensemble members use the same ocean model (NEMO, nucleus for European Modelling of the Ocean) and atmospheric forcing, there are differences in observational data and data assimilation techniques, as well as in the reanalysis initial states, NEMO versions, the sea-ice models, and physical parametrizations.
   The data assimilation methods differ in many points, including the deployed assimilation schemes which range from 3DVAR (three-dimensional variational data assimilation) to SEEK (Singular Evolutive Extended Kalman Filter). Furthermore, there are differences in the input observational dataset, in surface nudging, in the time-windows for assimilation and analysis as well as in the applied bias correction schemes. All those differences lead to an important dispersion between the reanalysis implementations and add up to the ensemble spread. (Storto et al., 2019)
   For further details we refer to Storto et al. (2019) as well as the individual data documentations, which are listed in the manuscript."

7. Line 105: The sentence starting "We also look into….ORCA025" appears out of place. Move up to line 72?

*We moved it up.*

8. Line 130: "additional area" needs to be clarified. Is this a catchment area?

   *Yes, it's a catchment area, we added the word "catchment".*

9. Section 3: Many math terms aren't defined clearly. E.g., $S\_A$, $S\_L$, $S\_O$, $F$. $\sigma^2\_k$. Make sure all terms are carefully defined.

   *We looked through them all and defined the terms that were missing.*

10. Line 142: Justify the neglect of atmospheric liquid water and ice.

    *We added the following justification:*

    *"Atmospheric liquid water and ice are neglected, as they represent only a very small fraction of water in comparison to atmospheric water vapor and lateral moisture fluxes – the liquid water portion stored in the atmosphere as well as liquid transports represent only about 1% compared to water vapor transports globally and even less in the northern latitudes (Hantel and Haimberger, 2016). Generally atmospheric water in liquid and solid phase are only significant in regions with high tropical cumulus clouds and over warm ocean currents (Serreze and Barry, 2014)."*

    *Hantel, Michael, and Haimberger, Leopold (2016). Grundkurs Klima. Berlin, Heidelberg: Springer Berlin Heidelberg.*

    *Serreze, M., and Barry, R. (2014). The Arctic Climate System (2nd ed., Cambridge Atmospheric and Space Science Series). Cambridge: Cambridge University Press. doi:10.1017/CBO9781139583817*

11. Line 151: What does A_total represent?  The Arctic Ocean?  The Arctic Ocean plus terrestrial catchments?

    *A_total is the sum of the Arctic Ocean and terrestrial areas, hence the total Arctic area considered in this study. We specified it in the paper.*

12. Line 158: What about groundwater contributions to the land water budget?  (And their changes in time).

    *That's a good point. As the ERA5 and ERA5-Land reanalyses do not contain groundwater storage and soil moisture is only given to the depth of 289cm. However, in the end we use GRACE satellite data to estimate land water storage changes, where groundwater changes are indeed included.* **Hence, we added the change of groundwater in equation 2 and also**

**mentioned the lack of groundwater in the reanalysis products in the data section:**
"The representation of frozen land components is not ideal and also groundwater storage is not represented in the examined reanalyses" (see also general comment 2 of Anonymous Referee #2)

As a side note, ECMWF currently works on increasing the number of soil layers and introducing a groundwater storage using a flexible, modular system called ECLand (Boussetta et al., 2021, Muñoz-Sabater et al., 2021).

Also, LISFLOOD, the hydrological model used in the GloFAS river discharge reanalysis, includes a groundwater module. This module consists of two reservoirs that store and subsequently output the water into the river channel after a certain time delay (Harrigan et al. 2020). We added this aspect in the data section as well.

New citations:
Muñoz-Sabater, J., Dutra, E., Agustí-Panareda, A., Albergel, C., Arduini, G., Balsamo, G., Boussetta, S., Choulga, M., Harrigan, S., Hersbach, H., Martens, B., Miralles, D. G., Piles, M., Rodríguez-Fernández, N. J., Zsoter, E., Buontempo, C., and Thépaut, J.-N.: ERA5-Land: a state-of-the-art global reanalysis dataset for land applications, Earth Syst. Sci. Data, 13, 4349–4383, https://doi.org/10.5194/essd-13-4349-2021, 2021.

Boussetta, S., Balsamo, G., Arduini, G., Dutra, E., McNorton, J., Choulga, M., Agustí-Panareda, A., Beljaars, A., Wedi, N., Muñoz Sabater, J., de Rosnay, P., Sandu, I., Hadade, I., Carver, G., Mazzetti, C., Prudhomme, C., Yamazaki, D., and Zsoter, E.: ECLand: The ECMWF Land Surface Modelling System, Atmosphere, 12, 723, https://doi.org/10.3390/atmos12060723, 2021.

13. Line 178: It says "we assume sea-ice to be transported by the ocean currents..." but sea ice moves (somewhat) independently from the surface ocean current. More explanation/justification is needed.

*That's also a very good point, that we investigated more thoroughly. In conclusion we changed our method of analysis as follows:*
*Instead of omitting sea ice transports we add them to our volume transports but at the same time remove the liquid water volume that is actually replaced by sea ice, which we call the equivalent liquid water flux.*

*Line 177 onwards now reads:*
"The liquid portion of F is calculated by integrating the cross-sectional velocity component along the side areas of the Arctic boundary. Additionally, we add ice transports, which are calculated analogical by integrating the cross-sectional ice velocity over the grid-point-average ice depth, multiplying it by the grid cells sea ice fraction and integrating it over the Arctic boundary. As

volume exchange between liquid ocean and sea-ice is conserved in the NEMO model, we additionally remove the liquid water volume that is actually replaced by sea ice, which we call the equivalent liquid water flux. The equivalent liquid water flux at a given grid point is calculated by integrating the liquid volume flux over the grid-point-average ice depth, multiplying it by the grid cells sea ice fraction and taking 90% of the result (as only ~90% of the icebergs are underwater). As ice velocities from the public CMEMS data portal are only available from two of the ocean reanalyses (ORAS5 and GLORYS2V4) we calculate the ice flux "correction" term for the GREP ensemble by taking the mean of only those two products. However, as the impact of the correction is quite similar for ORAS5 and GLORYS2V4 we believe that the correction is accurate enough for our purpose."

*This of course led to slight changes in figures 9-13 and tables 6 and 8, which have been updated in the revised manuscript. However, the results do not change substantially, and the main conclusions remain valid.*

14. Line 219: Equation (10) and the text about it are unclear.

    *We changed the equation and the text to the following:*

    "We calculate relative, decadal trends following Zsótér et al. (2020) and Stahl et al. (2012) by applying a linear regression to the annual mean time series:

    trend=10*slope/mean

    The *slope* of the time series is the annual trend obtained through the linear regression and the *mean* is the long-term annual mean of the timeseries. The ratio *slope/mean* is multiplied by the factor 10 to provide decadal trends. Hence a relative decadal trend of e.g. 0.1 is equal to an increase of 10% over a decade. All trends are calculated over the common period of the discharge datasets 1981-2019, except for GloFASE5L which is calculated over 1999-2018. We do not consider auto-correlation and determine significance using the Wald Test with a t-distribution, where p-values smaller than 0.05 are considered as significant."

15. Line 257: The sentence starting "Cuchi et al. (2020) run the hydrological model…." is out of place. Cut?

    Thanks for spotting this. We now cite the work of Cucchi et al. (2020) in the conclusion section. (see general comments)

16. Line 259: It says "Model runs with ERA5 forcing show similar river discharge seasonalities at the Lena catchment as GloFASERA5new." This isn't what I see in Figure 2 for the Lena. Check and cut or clarify.

> This was referring to the work of Cucchi et al. (2020) in the sentence above. Nevertheless, we cut it.

17. Line 278: "Again this could be caused by delayed river ice breakup and backwater that is considered in GRACE, but not in ERA5" is a bit misleading. GRACE observes the natural system, which includes delayed river ice breakup. The ERA5 model excludes does not represent these processes. Instead, maybe end the sentence with "that is observed by GRACE, but not considered in ERA5".

> Thanks for this comment. We clarified this accordingly.

18. Line 309: Say that the "hydrological analogy" means extrapolation to the un-gauged rivers and streams.

> We added this as well.

19. Figure 3: The legend identifies "GloFAS_{ERA5}" and similar, but the legends in Figure 2 call it "Glo_{E5}" (also Table 3). Use consistent notation throughout.

> We checked all notations and adapted them to be consistent.

20. Line 341: The sentence "Additionally Greenland features a storage decline of -134 km3 per year, accounting for roughly 50% of the total storage change" is unclear. Clarify and cite.

> *Clarification: The two sentences above address land water storage change in GRACE over the Pan-Arctic area excluding Greenland. In addition to this, Greenland features a storage decline of -134 km3 per year, also taken from GRACE.*

> *Correction of the passage:*
> "And just as for the four major basins, also for the sum of all Eurasian and North American catchments (excluding Greenland) GRACE data show a major decline of land water storage over the past decades, reaching -132 km3 per year for our area of interest, while land storage from ERA5 shows considerably smaller declines of -34 km3 per year. The largest changes for GRACE water storage occur over the Canadian Arctic Archipelago and the mountainous areas of Mackenzie and Yukon basin, suggesting a tight linkage to glacial melting. Additionally, GRACE water storage shows a strong decline of -134 km3 per year over the Greenlandic ice cap (north of Fram and Davis Strait), raising the total Pan-Arctic storage change to 266 km3 per year."

21. Figure 5: Remind the reader that the dashed lines sum to the brown line.

We added it to the figure caption.

22. Table 4: Add a column with units (applies to other tables too, trends in particular have an unclear unit). The "m3 s−1 * 10−3" unit contradicts Table 3.

    We corrected the units and added a column in the tables.

23. Section 4.2.3: The discussion on the ERA5 runoff glitches is useful. Speculate on how they could be fixed?

    *We added the following paragraph in section 4.2.3 (see also general questions):* "This discontinuity issue is not only limited to ERA5, but rather a general issue in reanalyses, as observation platforms are changing through time, making it practically very hard to make these products 100% homogeneous in time. Especially satellite data were not available in the early days and were introduced with the development and introduction of new instruments. If the redundancy is large enough, then any discontinuity impact should be less pronounced. However, specifically for snow there's only the IMS product that was introduced in 2004, and hence any inhomogeneity generates a larger impact. Thus, this impact could be reduced when using other data sets on top of the IMS product or instead of it, which ideally go further back in time."

24. Figure 9: Explain what the different lines in the right panel mean.

    Those are the different realizations of FW-input minus storage change using the various datasets described in the text. We clarified/repeated it in the figure caption.

25. Line 512: It says "both F and runoff R feature adjustments beyond their a priori spreads, demonstrating that the a priori uncertainties are larger than indicated as systematic biases are not incorporated". What are the likely systematic biases?

    *We added the following explanation after line 513:*

    "The state-of-the art reanalyses exhibit systematic errors concerning the runoff seasonality, where the seasonal runoff peaks in summer are too low in comparison to observations, while winter and spring values are too high. Due to the lack of reliable seasonal observations of the oceanic volume fluxes, it's hard to define systematic biases in the ocean reanalyses. However, all four ocean reanalyses feature quite low September volume fluxes, which are not found in their forcing components (see figure 9). Uotila et al. (2019) assess ten ocean reanalyses, including CGLORS, FOAM, GLORYS and ORAS5, specifically in the polar regions and find multiple systematic errors concerning sea ice thickness and extent, temperature profiles, mixed layers as well as ocean

transports. Unfortunately, seasonal cycles of volume transports were not assessed either, however seasonal cycles of sea ice components and heat transports did exhibit systematical biases. Further analyses would be necessary to find certain, systematical errors in the seasonal volume transports."

New reference:
Uotila, P., Goosse, H., Haines, K. et al. An assessment of ten ocean reanalyses in the polar regions. Clim Dyn 52, 1613–1650 (2019). https://doi.org/10.1007/s00382-018-4242-z

26. Line 535: State briefly the origin of the ERA5-Land runoff declines of 5-6% and comment on their realism.

*We added the following explanation:*

"Those declines in ERA5-L runoff are caused by similar declines in P-E from ERA5 short-term forecasts, as P-E is used as a forcing in ERA5-L (see figure 6/table 4). As observations agree on an increase of river discharge, those declines are deemed unrealistic. An improvement may possibly be achieved when taking the divergence of moisture flux (VIWVD) as forcing, as VIWVD, which is computed from analysed fields rather than short-term forecasts, features similar trends as discharge observations. See J. Mayer et al. (2021) for an assessment of the ERA5 water budget and the different behavior of budget quantities taken from analyses or short-term model forecasts."

New citation:
Mayer, J., Mayer, M., & Haimberger, L. (2021). Consistency and Homogeneity of Atmospheric Energy, Moisture, and Mass Budgets in ERA5, Journal of Climate, 34(10), 3955-3974, https://doi.org/10.1175/JCLI-D-20-0676.1.

27. Line 552: It says "With oceanic and land storage declining...", yet many papers exist on the accumulation of freshwater in the western Arctic Ocean (e.g., see Proshutinsky et al., 2019, 10.1029/2019JC015281). Mention and comment on this issue.

*This is not necessarily contradictory to our studies as GRACE measures changes in **mass** not freshwater. Furthermore, considered timespan and areal extend have great effects, as storage is tightly linked with winds and circulation patterns. Concerning the spatial distribution of the trends, we examined trend maps and also found slight increases of mass in the Chukchi and Beaufort Sea (= western Arctic), however stronger declines in the eastern part of the Arctic, with the strongest changes in Baffin Bay.*
*We added the following paragraph after line 507:*

"Other papers mention the accumulation of freshwater in the western Arctic Ocean (e.g, Proshutinsky et al., 2019). This is not necessarily contradictory to our findings of a slight decline in ocean storage, as GRACE does not measure the accumulation/decline of freshwater but rather the change in mass. Furthermore, areal differences (we consider the whole Arctic Ocean) and differences in the considered timespan have great effects, as storage is tightly linked with winds and circulation patterns and exhibits strong nonseasonal fluctuations. Longer time series would be needed to determine whether the trends that we found are caused primarily by such fluctuations or indicate a true loss of mass."

Technical Corrections:

1. Many places: Apostrophes are not used correctly and there are spelling and grammar errors.

   We revised the whole paper for correct grammar and to eliminate spelling errors.

2. Abstract, line 16: Reword "look into Greenlandic discharge"

   Reworded to:
   In addition we examine Greenlandic discharge,...

3. Abstract, line 22: Which "data-sets"?  Be specific for clarity.

   As the systematic biases come from the reanalysis and ocean-reanalysis datasets we added those.

4. Line 169: "reference salinity" should read "reference density" I think.

   That's correct, we changed it.

5. Figure 2: What are R and \mu in Figure 2 legends?

   R is runoff and \mu is the long-term mean. We added them to the figure caption.

---

## Author Comment (AC2)

**Response to Anonymous Referee #2**

Thank you very much for your constructive feedback and your positive comments, you addressed some important points. Your clarifications helped to make the manuscript clearer for the reader. Our responses are provided in green together with your original comments in black.
We really appreciate your time and insight in reviewing our manuscript!

Kind regards,

Susanna (on behalf of all co-authors)

General comments:

The manuscript is well organized, but there are some spelling and grammatical errors that needs to be considered, including the use of commas and apostrophes. I also suggest to avoid using words such as "spurious", "huge", "clearly" etc., especially for the results and conclusions sections (see also specific comments).

We revised the whole paper and correct spelling and grammatical errors and we made sure to avoid words as "spurious", "huge", "clearly",...

Considering that previous studies focusing on the Arctic drainage basin have used different approaches and motivations for its geographical domain, I am missing a motivation for the chosen boundary of the Arctic Ocean drainage basin in this study, and why e.g., Hudson Bay, and James Bay was not included? (e.g., L302-305).

*We added a clarification at line 121:*
"Figure 1 presents the study domain. As there's no strict boundary to the south, the definition of the Arctics geographic extent varies between past studies and there's no general rule whether to include Greenland and the Hudson Bay or not. We chose our study domain to be consistent with Tsubouchi et al. (2012) as we wanted to compare the oceanic fluxes from ocean reanalysis with the observation-based estimates from the ARCGATE project. The Arctic Ocean is bounded by the position of hydro-graphic moorings in the main gateways. ..."

Clarify also in L313 that total drainage area refers to the area for this study.

We clarified this.

How do these reanalysis products take frozen components of the freshwater system into consideration, e.g., glaciers and permafrost, considering that many of the river basins in the study are underlain by permafrost? For example, lines

497-498 includes an interesting aspect that I would like to see more elaboration on.

*We added an explanation on the representation of frozen land components in reanalyses in the data section (section 2):*
"The representation of frozen land components is not ideal and also groundwater storage is not represented in the examined reanalyses. In HTESSEL, the land-surface model used in ERA5 and ERA5-Land, glaciers are represented as large amounts of snow which are kept fixed to 10 m of snow water equivalent. When melting conditions are reached, the snow produces a water influx to the soil and consequently contributes to the total runoff. However, the mass balance is not accounted for over glaciers as the snow is restocked to constantly stay at the fixed 10m level and hence changes in the glacial storage component can't be assessed properly. The soil water content includes liquid as well as frozen components and thus also includes permafrost. When the soil temperature reaches melting conditions, the soil water contributes to sub-surface runoff and the soil water storage declines. However, a recent study by Cao et al. (2020) concluded that ERA5L soil data are not optimal for permafrost research, due to a warm bias in soil temperature that leads to an overestimation of the active-layer thickness and an underestimation of the near-surface permafrost area. Therefore, we additionally use GRACE (Gravity Recovery & Climate Experiment) satellite data for our final analyses to accurately estimate the land water storage, as GRACE includes liquid water (including groundwater), glaciers and permafrost."

*Additionally, we justified the use of ERA5 land water storage for certain purposes at lines 497-498:*
"Oceanic transports out of the Arctic domain exceed the atmospheric moisture entering the Arctic (6295±121km3) by nearly 5%, indicating an annual loss of water volume of roughly 300km3. The bulk part of this loss is generated through terrestrial water mass losses. Even though the representation of frozen land components is not ideal in the considered reanalyses, the comparison of GRACE mass changes to the sum of ERA5 storage changes (snow and soil water) and glacial changes taken from literature agreed well. Therefore, we use land storage changes in ERA5 (excluding glaciers) to estimate which terrestrial sources contributed to what extend to the general storage decline in the Arctic. We found that approximately 50% of the 266 km3/year decline are generated through liquid and solid discharge from Greenland, while about 40% come from Arctic glaciers (excluding Greenland) and the remaining 10% are the result of a decline in land water storage due to permafrost and snow cover degradation."

New references:
Cao, B., Gruber, S., Zheng, D., and Li, X.: The ERA5-Land soil temperature bias in

permafrost regions, The Cryosphere, 14, 2581–2595, https://doi.org/10.5194/tc-14-2581-2020, 2020.

In the conclusions, I am missing a general discussion on implications for future studies and assessments of freshwater budgets of the Arctic Ocean.

*This question is similar to the 2. general comment by Anonymous Referee #1. We revised the whole conclusion section. Some of the key points that were added are the following:*
"Summarizing we refined past Arctic water budget estimates (Serreze et al., 2006; Dickson et al., 2007, e.g.,) and their uncertainties by combining some of the most recent reanalyses data-sets and observations, and by applying a variational optimization scheme. The variational adjustment worked very well on an annual scale and brought reliable estimates of the volume budget terms, requiring only moderate adjustments of less than 3% for each individual term. Adjustments are considered reliable if budget closure was achievable within the respective terms error bounds and if the terms were comparable to estimates from past studies."

"Especially when calculating Pan-Arctic runoff, caution is needed. Our results show that seasonal peaks of river discharge are underestimated in almost all of the assessed reanalyses (ERA5, ERA5-Land, GloFASE5, GloFASE5L). The biggest anomalies are caused due to inhomogeneities in the data assimilation system (ERA5 and GloFASE5) which led to a great underestimation of runoff, especially at the latter half of the time series. However also reanalyses without data assimilation (ERA5-Land andGloFASE5L) were not able to reproduce the seasonal cycle of river discharge accurately. On the other hand, we find distinct improvements in the new GloFASERA5newproduct, especially when investigating seasonal cycles and long term means it features vast enhancements compared to its precursors."

"When extrapolating observed river discharge to the whole Pan-Arctic area we found that the common method of hydrological analogy tends to underestimate the discharge peaks. We therefore advise to use river discharge observations where available and reliable runoff/discharge estimates from reanalyses (e.g., GloFASERA5new or ERA5-Land) to extrapolate discharge to the ungauged areas."

"To further refine the budget estimates, longer time scales of all budget terms would be needed. For example, one could repeat the analysis using the back extension of ERA5 which goes back to 1950. There's also a new bias corrected ERA5 data set (Cucchi et al., 2020, WFDE5,), that could be examined in relation to the Arctic water budget. Further it would help to include a precipitation observation data set, preferably one that combines available satellite-based and

gauge-based data sets.
Concerning the biases in ocean reanalyses, one could refer to oceanographic data for comparison. Generally, comparison to oceanographic data is difficult, as observations are limited concerning their temporal and spatial coverage. Nevertheless, the unique form of the Arctic Ocean (as water leaves and enters only through a handful of gateways) allows relatively easy measurements of the in- and outgoing fluxes. This was done in the ARCGATE project, where data from arrays of moored instruments (like e.g., Acoustic Doppler Current Profilers, MicroCAT – CTD Sensors and Seagliders) were taken to estimate transports through the Arctic gateways. Our results showed that it is possible to estimate annual fluxes into and out of the Arctic Ocean quite accurately, however the moored instruments did not measure the velocity field accurately enough to resolve the barotropic wave signal arising from temporally varying runoff (T. Tsubouchi, personal communication 2021) leading to differences in the seasonality of the fluxes. A longer measuring period with an even denser monitoring network could help with this aspect."

Specific comments:

L12: I suggest to avoid the use of "spurious" and instead explain or reference to what you are referring to.

Rephrased to:
Seasonal river discharge peaks are underestimated in ERA5 and GloFAS v2.1 by up to 50%, due to pronounced declining runoff trends which are caused by two temporal inhomogeneities in ERA5.

L37: consider removing "remarkably"

We removed it.

L41-43: Consider rephrasing for clarity and also specify the part on climatological conditions.

Rephrased to:
In addition, significant portions of the rivers discharge may bypass the gauging stations through braided channels or as submarine groundwater. Furthermore, also climatological conditions pose a hindrance to gauge measurements, as the low temperatures in the northern latitudes often lead to river freeze up in late autumn and flooding in spring due to river-ice break up (Syed et al., 2007).

L45: avoid using "huge"

We changed it to "great".

L47-48: Consider rephrasing for clarity. and
L48-49: This is not very clear, please explain what you mean by "spurious" (see also previous comment related to this).

We rephrased it to:
However data assimilation systems can introduce biases, as temporal changes in the observing system are inevitable and may lead to inhomogeneities in the time series. One known change is the introduction of the IMS (Interactive Multisensor Snow and Ice Mapping System) snow product in ERA5, which led to a negative shift in ERA5's snowmelt and consequently also runoff. (Hersbach et al., 2020, Zsoter et al., 2020)

L92: Which 16 rivers were included in the study, and how was the shorter observational records treated for the analysis in comparison to the longer observational records?

We added the names of the 16 rivers:
Pur, Taz, Khatanga, Anabar, Olenek, Yana, Indigirka, Alazeya, Abadyr, Kobuk, Hayes, Tana, Tuloma, Ponoy, Onega, Mezen

And clarified the consideration of shorter vs longer records:
"We calculated an observation-based Pan-Arctic river discharge for the period of 1979 to 2019. Therefore, we calculated discharge for every time step (= every month) separately and used all river discharge measurements that were available at this certain timestamp. Analogous river discharge for the ungauged area was estimated for each individual timestamp over the area that was not gauged at this time, using two calculation methods (see section 4.2)."

L119-120: What about frozen storage components, such as glaciers?

We added permafrost and glaciers as those were considered too through GRACE. (See answer of general comments)

L134: I suggest to add references to earlier studies, and revise "popular" to "common" – if this is what you are referring to?

Yes, we changed the wording and added a couple references:
A common way to calculate the oceanic freshwater budget is through the assumption of a reference salinity (e.g., Serreze et al., 2006; Dickson et al., 2007; Curry et al., 2011; Haine et al., 2015).

I suggest to remove a, b, c in subheadings (e.g., L196, 205).

We removed those.

L244: consider removing "clearly"

We removed it.

Fig 4: Is this figure only considering the shorter time series of the 16 catchments, or for the full time period (1981-2019)? Same question for figure 6 and the observed Pan-Arctic river discharge data.

The black lines (observations) of Figure 4 consider the shorter time series of all catchments, while the reanalyses are taken over the full time period. We considered the difference as insignificant, as the purpose of the figure was just to show the differences between the different extrapolation methods.
We added all time periods into the figure captions.

L497-498: This is an interesting aspect that I would like to see more elaboration on.

See answer to general comments

L527: I suggest to include references here, and do you mean "common" rather than "popular"?

Yes, we again changed popular to common and added references (e.g., Shiklomanov and Shiklomanov, 2003).

L528: How does this result compare to other studies?

*We a comparison to other studies:*
"We estimate Pan-Arctic river discharge from gauge observations using monthly correction factors from GloFASERA5new, as the popular method of hydrological analogy tends to underestimate the high flow summer peaks (see Fig. 4) and obtain a long-term annual flux of 4031km3 ± 203 (excluding Greenland). To compare our results to past studies we adapted the time periods and areal extents accordingly and found reasonable accordance with Haine et al. (2015), who combined runoff from ERA-Interim with river discharge observations and obtained a total discharge about 5% higher than our estimate. An even better agreement was found with the estimates made by Shiklomanov et al. (2021a), as the total Pan-Arctic discharges (including Greenland) agree within 2%."

L529: Runoff from ERA5 is substantially "too low" – do you mean "underestimated compared to observed discharge" or similar?

Yes, that's what we meant. We rephrased it accordingly.

L531: Please consider rephrasing and describe the "unrealistic" aspects.

*We rephrased it:*
"Those strong declines are caused by two inhomogeneities (1992 and 2004) in ERA5's snow melt time-series and contradict the discharge increases found in gauge observations. Those inhomogeneities are caused by a loss of snow through changes in the data assimilation system."

L548: What is considered "trustworthy" here – please explain.

*By trustworthy we mean the data products we have most confidence in after comparing them amongst each other and to observational datasets. We changed the sentence to the following:*
"Comparing the estimates of freshwater input into the Arctic Ocean that we have most confidence in after the preceding analysis (listed in table 7), to volume transports from ocean reanalysis yields..."

L555: What is considered "reliable" – please explain

*We added an explanation:*
Adjustments are considered reliable if budget closure was achievable within the respective terms error bounds and if the terms were comparable to estimates from past studies.

L559: What would be a full success here, please elaborate.

*We changed the sentence to the following:*
On a seasonal scale however, the adjustment process was not a full success, as some of the adapted fluxes fell out of their a priori uncertainty range. A full success would include elimination of the budget residuals for every single month, while at the same time staying inside the respective a priori spreads of the individual terms. This is very likely caused by systematic errors being present in the datasets, or at least in their seasonal cycles, that are not taken into account in our a priori uncertainty estimates.

L560: revise month to months

We changed it.

L571: Please specify what you refer to with "in most reanalyses".
L572: Please specify what you refer to with "spurious signals".

*We changed it to:*
"Our results show that seasonal peaks of river discharge are underestimated in almost all of the assessed reanalyses (ERA5, ERA5-Land, GloFASE5, GloFASE5L).

The biggest anomalies are caused due to inhomogeneities in the data assimilation system (ERA5 and GloFASE5) which led to a great underestimation of runoff, especially at the latter half of the time series. However also reanalyses without data assimilation (ERA5-Land and GloFASE5L) were not able to reproduce the seasonal cycle of river discharge accurately."

---

## Author Response (AR1)

**Response to Anonymous Referee #1**

General/overarching comments:

*To answer your general comments, we revised the whole conclusion section (from line 555 onward). See changes in the marked-up manuscript.*

1. Although river discharge data and land/ocean water storage data are used carefully, the paper doesn't use oceanographic data to estimate the marine water fluxes (it only uses reanalysis data). This isn't a major problem because the paper focuses on runoff and river discharge, but it should be mentioned and discussed somewhere (Conclusions?).

   *We added the following paragraph to the conclusions section:*

   Concerning estimation of biases in ocean reanalyses, one could in principle draw on information from oceanographic data for comparison. A main difficulty with oceanographic data is the generally limited temporal and spatial coverage. Nevertheless, the unique form of the Arctic Ocean (as water leaves and enters only through a handful of gateways) allows relatively easy measurements of the in- and outgoing fluxes. As an example, measurements from arrays of moored instruments (like e.g., Acoustic Doppler Current Profilers, MicroCAT – CTD Sensors and Seagliders) were taken to estimate transports through the Arctic gateways using a mass-consistent framework (Tsubouchi et al., 2012, 2018). Our results however showed that the moored instruments did not measure the velocity field accurately enough to resolve the barotropic wave signal arising from temporally varying runoff (Tsubouchi, 2019) leading to errors in the seasonality of the net volume flux. A longer measuring period with an even denser monitoring network could help with this aspect.

2. Related to point 1: What's the scope/opportunity for future improvements on the Arctic water budget analysis? What model and data assimilation improvements would help? What data are needed to refine the budget estimates? Again, this isn't a major omission, but it will help set the context for future work if this point is discussed somewhere (Conclusions?).

   What model and data assimilation improvements would help?
   *We added the following paragraph in section 4.2.3:*
   This discontinuity issue is not only limited to ERA5, but rather a general issue in reanalyses, as observation platforms are changing through time, making it practically very hard to make these products perfectly homogeneous in time. Especially satellite data were not available in the early days and were introduced with the development and introduction of new instruments. If the

redundancy is large enough, then any discontinuity impact should be less pronounced. However, specifically for snow there is only the IMS product that was introduced in 2004, and hence any inhomogeneity generates a larger impact. Thus, this impact could possibly be reduced when using other data sets on top of the IMS product or instead of it, which ideally go further back in time.

What's the scope/opportunity for future improvements on the Arctic water budget analysis? What data are needed to refine the budget estimates?

*We added some aspects in the conclusion section, e.g.:*
To further refine the budget estimates, longer time series of all budget terms would be needed. For example, one could repeat the analysis using the back extension of ERA5 which goes back to 1950. There is also a new bias-corrected ERA5 data set (WFDE5, Cucchi et al., 2020), that could be examined in terms of the Arctic water budget. Further it would help to include a precipitation observation data set, preferably one that combines available satellite-based and gauge-based data sets.

Specific Comments:

1. Line 50: Cite where it says the ERA5 runoff features spurious trends.

   We added a citation of Zsoter et al. (2020)

2. Figures 1 and 2: What is the source of catchment data in Figures 1 and 2?

   We added the following catchment sources (shapefiles):

   **Individual river catchment outlines** were taken from the CEO Water Mandate Interactive Database of the World's River Basins (http://riverbasins.wateractionhub.org/)
   **Regional outlines** (CAA etc.): GRDC (2020): WMO Basins and Sub-Basins / Global Runoff Data Centre, GRDC. 3rd, rev. ext. ed. Koblenz, Germany: Federal Institute of Hydrology (BfG).

3. Section 2: Add a table containing information on the runoff and discharge sources (ERA5, ERA5-Land, GloFAS…, GREP etc.)

   We added a table containing all runoff and river discharge sources and key information of the individual products.

| Product | Description | Variable | Period | |
|---|---|---|---|---|

| ERA5 | Fifth generation ECMWF reanalysis using IFS (+ HTESSEL) | Runoff [m/s] | 1979-2019 (back extension to 1950 available) | Hersbach et al., 2020 |
|---|---|---|---|---|
| ERA5-Land | Offline simulation of ERA5 without DA using HTESSEL | Runoff [m/s] | 1981-2019 (back extension to 1950 expected in 2021) | Muñoz-Sabater et al., 2021 |
| GloFAS$_{E5}$ | ERA5 runoff + simplified LISFLOOD | River discharge [m$^3$/s] | 1979-2019 | Harrigan et al., 2019 |
| GloFAS$_{E5L}$ | ERA5-Land runoff + simplified LISFLOOD | River discharge [m$^3$/s] | 1999-2018 | - |
| GloFAS$_{E5new}$ | Full configuration of LISFLOOD | River discharge [m$^3$/s] | 1979-2019 | - |
| Bt06 | Runoff climatology used in ORCA025 | River discharge [m$^3$/s] | Climatology | Bourdalle-Badie and Treguier, 2006 |
| Observations | Measurements at gauging stations | River discharge [m$^3$/s] | - | - |

New citation:
Muñoz-Sabater, J., Dutra, E., Agustí-Panareda, A., Albergel, C., Arduini, G., Balsamo, G., Boussetta, S., Choulga, M., Harrigan, S., Hersbach, H., Martens, B., Miralles, D. G., Piles, M., Rodríguez-Fernández, N. J., Zsoter, E., Buontempo, C., and Thépaut, J.-N.: ERA5-Land: a state-of-the-art global reanalysis dataset for land applications, Earth Syst. Sci. Data, 13, 4349–4383, https://doi.org/10.5194/essd-13-4349-2021, 2021.

4. Line 70: "river discharge" includes both liquid water and ice (presumably)?

   Yes, it does, we clarified it.

5. Line 73: For clarity, say that "associated domain" means the catchment area.

   We clarified that.

6. Line 104: It talks about "different bulk formulas and differences in the data assimilation…"  Different to what?  Be specific.

*We specified some of the differences and referred to the individual documentations for further details. The following paragraph was added after line 103:*

While the GREP ensemble members use the same ocean modeling core and atmospheric forcing (ERA-Interim, Dee et al., 2011), there are differences in used observational data and data assimilation techniques, as well as in the reanalysis initial states, NEMO (Nucleus for European Modelling of the Ocean) versions, the sea-ice models, physical and numerical parametrizations, and air-sea flux formulations. The data assimilation methods differ in many points, including the deployed assimilation schemes which range from 3DVAR (three-dimensional variational data assimilation) to SEEK (Singular Evolutive Extended Kalman Filter). Furthermore, there are differences in the input observational dataset, in surface nudging, in the time-windows for assimilation and analysis as well as in the applied bias correction schemes. All those differences lead to an important dispersion between the reanalysis implementations and add up to the ensemble spread (Storto et al., 2019). For further details we refer to Storto et al. (2019) as well as the individual data documentations.

7. Line 105: The sentence starting "We also look into....ORCA025" appears out of place. Move up to line 72?

   We moved it up.

8. Line 130: "additional area" needs to be clarified. Is this a catchment area?

   Yes, it's a catchment area, we added the word "catchment".

9. Section 3: Many math terms aren't defined clearly. E.g., $S_A$, $S_L$, $S_O$, F. $\sigma^2_k$. Make sure all terms are carefully defined.

   We looked through them all and defined the terms that were missing.

10. Line 142: Justify the neglect of atmospheric liquid water and ice.

    *We added the following justification:*

    Atmospheric liquid water and ice are neglected, as they represent only a very small fraction of water in comparison to atmospheric water vapor and lateral moisture fluxes. Generally atmospheric water in liquid and solid phase are only present in significant amounts in regions with high tropical cumulus clouds and over warm ocean currents (Serreze and Barry, 2014).

11. Line 151: What does $A_{total}$ represent? The Arctic Ocean? The Arctic Ocean plus terrestrial catchments?

A_total is the sum of the Arctic Ocean and terrestrial areas, hence the total Arctic area considered in this study. We specified it in the paper.

12. Line 158: What about groundwater contributions to the land water budget? (And their changes in time).

*That's a good point. As the ERA5 and ERA5-Land reanalyses do not contain groundwater storage and soil moisture is only given to the depth of 289cm. However, in the end we use GRACE satellite data to estimate land water storage changes, where groundwater changes are indeed included.*
***Hence, we added the change of groundwater in equation 2 and also mentioned the lack of groundwater in the reanalysis products in the data section:***

"Groundwater storage is not represented in ERA5 and ERA5-Land and also the representation of frozen land components is not ideal in HTESSEL, as glaciers are depicted as large amounts of snow which are kept fixed to 10 m of snow water equivalent. When melting conditions are reached, the snow produces a water influx to the soil and consequently contributes to the total runoff. However, the mass balance is not accounted for over glaciers as the snow is restocked to constantly stay at the fixed 10 m level and hence changes in the glacial storage component cannot be assessed properly. The soil water content includes liquid as well as frozen components and thus also includes permafrost. When the soil temperature reaches melting conditions, the soil water contributes to sub-surface runoff and the soil water storage declines. However, a recent study by Cao et al. (2020) concluded that ERA5-Land soil data are not optimal for permafrost research, due to a warm bias in soil temperature that leads to an overestimation of the active-layer thickness and an underestimation of the near-surface permafrost area. Therefore, we additionally include GRACE (Gravity Recovery and Climate Experiment) satellite data, as land water storage from GRACE includes changes in soil moisture (including permafrost), glaciers, snow, surface water, aquifers and groundwater."

*As a side note, ECMWF currently works on increasing the number of soil layers and introducing a groundwater storage using a flexible, modular system called ECLand (Boussetta et al., 2021, Muñoz-Sabater et al., 2021).*

*Also, LISFLOOD, the hydrological model used in the GloFAS river discharge reanalysis, includes a groundwater module. This module consists of two reservoirs that store and subsequently output the water into the river channel after a certain time delay (Harrigan et al. 2020). We added this aspect in the data section as well.*

New citations:

Muñoz-Sabater, J., Dutra, E., Agustí-Panareda, A., Albergel, C., Arduini, G., Balsamo, G., Boussetta, S., Choulga, M., Harrigan, S., Hersbach, H., Martens, B., Miralles, D. G., Piles, M., Rodríguez-Fernández, N. J., Zsoter, E., Buontempo, C., and Thépaut, J.-N.: ERA5-Land: a state-of-the-art global reanalysis dataset for land applications, Earth Syst. Sci. Data, 13, 4349–4383, https://doi.org/10.5194/essd-13-4349-2021, 2021.

Boussetta, S., Balsamo, G., Arduini, G., Dutra, E., McNorton, J., Choulga, M., Agustí-Panareda, A., Beljaars, A., Wedi, N., Muñoz Sabater, J., de Rosnay, P., Sandu, I., Hadade, I., Carver, G., Mazzetti, C., Prudhomme, C., Yamazaki, D., and Zsoter, E.: ECLand: The ECMWF Land Surface Modelling System, Atmosphere, 12, 723, https://doi.org/10.3390/atmos12060723, 2021.

13. Line 178: It says "we assume sea-ice to be transported by the ocean currents..." but sea ice moves (somewhat) independently from the surface ocean current. More explanation/justification is needed.

   *That's also a very good point, that we investigated more thoroughly. In conclusion we changed our method of analysis as follows:*
   *Instead of omitting sea ice transports we add them to our volume transports but at the same time remove the liquid water volume that is actually replaced by sea ice, which we call the equivalent liquid water flux.*

   *Line 177 onwards now reads:*
   The liquid portion of F is calculated by integrating the cross-sectional velocity component along the side areas of the Arctic boundary. Additionally, we add ice transports, which are calculated analogously by integrating the cross-sectional ice velocity over the grid-point-average ice depth and integrating it over the Arctic boundary. As volume exchange between liquid ocean and sea-ice is conserved in the NEMO model, we additionally remove the liquid water volume that is actually replaced by sea ice, which we call the equivalent liquid water flux. The equivalent liquid water flux at a given grid point is calculated by integrating the liquid volume flux over the grid-point-average ice depth and taking 90% of the result (as only 90% of the icebergs are underwater). As ice velocities from the public CMEMS data portal are only available from two of the ocean reanalyses (ORAS5 and GLORYS2V4) we calculate the ice flux "correction" term for the GREP ensemble by taking the mean of those two products. However, as the impact of the correction is quite similar for ORAS5 and GLORYS2V4 we believe that the correction is accurate enough for the purpose of this study.

   *This of course led to slight changes in figures 9-13 and tables 6 and 8, which have been updated in the revised manuscript. However, the results do not change substantially, and the main conclusions remain valid.*

14. Line 219: Equation (10) and the text about it are unclear.

    *We changed the equation and the text to the following:*

    "We calculate relative, decadal trends following Zsótér et al. (2020) and Stahl et al. (2012) by applying a linear regression to the annual mean time series:

    trend=10*slope/mean

    The *slope* of the time series is the annual trend obtained through the linear regression and the *mean* is the long-term annual mean of the timeseries. The multiplication factor 10 results as we calculate trends over a fixed 10-year period. Hence a trend of e.g., 0.1 is equal to an increase of 10% over a decade. All trends are calculated over the common period of the discharge datasets 1981-2019, except for GloFASE5L which is calculated over 1999-2018.We do not consider temporal auto-correlation, assuming that subsequent annual means are only weakly correlated, and determine significance using the Wald Test with a t-distribution, where p-values smaller than 0.05 are considered as significant.

15. Line 257: The sentence starting "Cuchi et al. (2020) run the hydrological model...." is out of place. Cut?

    Thanks for spotting this. We now cite the work of Cucchi et al. (2020) in the conclusion section. (see general comments)

16. Line 259: It says "Model runs with ERA5 forcing show similar river discharge seasonalities at the Lena catchment as GloFASERA5new." This isn't what I see in Figure 2 for the Lena. Check and cut or clarify.

    This was referring to the work of Cucchi et al. (2020) in the sentence above. Nevertheless, we cut it.

17. Line 278: "Again this could be caused by delayed river ice breakup and backwater that is considered in GRACE, but not in ERA5" is a bit misleading. GRACE observes the natural system, which includes delayed river ice breakup. The ERA5 model excludes does not represent these processes. Instead, maybe end the sentence with "that is observed by GRACE, but not considered in ERA5".

    Thanks for this comment. We clarified this accordingly.

18. Line 309: Say that the "hydrological analogy" means extrapolation to the un-gauged rivers and streams.

We added this as well.

19. Figure 3: The legend identifies "GloFAS_{ERA5}" and similar, but the legends in Figure 2 call it "Glo_{E5}" (also Table 3). Use consistent notation throughout.

We checked all notations and adapted them to be consistent.

20. Line 341: The sentence "Additionally Greenland features a storage decline of -134 km3 per year, accounting for roughly 50% of the total storage change" is unclear. Clarify and cite.

*Clarification: The two sentences above address land water storage change in GRACE over the Pan-Arctic area excluding Greenland. In addition to this, Greenland features a storage decline of -134 km3 per year, also taken from GRACE.*

*Correction of the passage:*
And just as for the four major basins, also for the sum of all Eurasian and North American catchments (excluding Greenland) GRACE data show a major decline of land water storage over the past decades, reaching -132 km3 per year for our area of interest, while land storage from ERA5 shows considerably smaller declines of -34 km3 per year. The largest changes for GRACE water storage occur over the Canadian Arctic Archipelago and the mountainous areas of Mackenzie and Yukon basin (maps not shown), suggesting a tight linkage to glacial melting. Additionally, GRACE water storage shows a strong decline of -134 km3 per year over the Greenlandic ice cap north of Fram and Davis Strait, raising the total Pan-Arctic storage change to 266 km3 per year.

21. Figure 5: Remind the reader that the dashed lines sum to the brown line.

We added it to the figure caption.

22. Table 4: Add a column with units (applies to other tables too, trends in particular have an unclear unit). The "m3 s−1 ∗ 10−3" unit contradicts Table 3.

We corrected the units and added a column in the tables.

23. Section 4.2.3: The discussion on the ERA5 runoff glitches is useful. Speculate on how they could be fixed?

*We added the following paragraph in section 4.2.3 (see also general questions):*
This discontinuity issue is not only limited to ERA5, but rather a general issue in reanalyses, as observation platforms are changing through time, making it practically very hard to make these products perfectly homogeneous in time.

Especially satellite data were not available in the early days and were introduced with the development and introduction of new instruments. If the redundancy is large enough, then any discontinuity impact should be less pronounced. However, specifically for snow there is only the IMS product that was introduced in 2004, and hence any inhomogeneity generates a larger impact. Thus, this impact could possibly be reduced when using other data sets on top of the IMS product or instead of it, which ideally go further back in time.

24. Figure 9: Explain what the different lines in the right panel mean.

Those are the different realizations of FW-input minus storage change using the various datasets described in the text. We clarified/repeated it in the figure caption.

25. Line 512: It says "both F and runoff R feature adjustments beyond their a priori spreads, demonstrating that the a priori uncertainties are larger than indicated as systematic biases are not incorporated". What are the likely systematic biases?

*We added the following explanation after line 513:*
The state-of-the art reanalyses exhibit systematic errors in their runoff seasonalities, as the seasonal runoff peaks in summer are too low in comparison to observations, while winter and spring values are too high. Due to the lack of reliable seasonal observations of the oceanic volume fluxes, it is hard to define systematic biases in the ocean reanalyses. However, all four ocean reanalyses feature quite low September volume fluxes, which are not found in their forcing components (see figure 9). Uotila et al. (2019) assess ten ocean reanalyses, including CGLORS, FOAM, GLORYS and ORAS5, specifically in the polar regions and find multiple systematic errors concerning sea ice thickness and extent, temperature profiles, mixed layers as well as ocean transports. Seasonal cycles of volume transports were not assessed, however seasonal cycles of sea ice components and heat transports did exhibit systematic errors. Further analyses would be necessary to come up with robust estimates of the bias in seasonal volume transports.

New reference:
Uotila, P., Goosse, H., Haines, K. et al. An assessment of ten ocean reanalyses in the polar regions. Clim Dyn 52, 1613–1650 (2019). https://doi.org/10.1007/s00382-018-4242-z

26. Line 535: State briefly the origin of the ERA5-Land runoff declines of 5-6% and comment on their realism.

*We added the following explanation:*

These declines in ERA5-L runoff are caused by similar declines in P-E from ERA5, as P-E is used as a forcing in ERA5-L (see figure 6 and table 5). As observations agree on an increase of river discharge, these declines are deemed unrealistic. An improvement may possibly be achieved when taking the divergence of moisture flux (VIWVD) as forcing, as VIWVD, which is computed from analysed fields rather than short-term forecasts, features similar trends as discharge observations.

27. Line 552: It says "With oceanic and land storage declining...", yet many papers exist on the accumulation of freshwater in the western Arctic Ocean (e.g., see Proshutinsky et al., 2019, 10.1029/2019JC015281). Mention and comment on this issue.

    *This is not necessarily contradictory to our studies as GRACE measures changes in* **mass** *not freshwater. Furthermore, considered timespan and areal extend have great effects, as storage is tightly linked with winds and circulation patterns. Concerning the spatial distribution of the trends, we examined trend maps and also found slight increases of mass in the Chukchi and Beaufort Sea (= western Arctic), however stronger declines in the eastern part of the Arctic, with the strongest changes in Baffin Bay.*
    *We added the following paragraph after line 507:*
    Other papers mention the accumulation of freshwater in the western Arctic Ocean (e.g. Proshutinsky et al., 2019). This is not necessarily contradictory to our findings of a slight decline in ocean storage, as GRACE does not measure the accumulation/decline of freshwater but rather the change in mass. Furthermore, areal differences (we consider the whole Arctic Ocean) and differences in the considered timespan have strong effects, as storage is tightly linked with winds and circulation patterns and exhibits strong nonseasonal fluctuations. Longer time series would be needed to determine whether the trends that we found are caused primarily by such fluctuations or indicate a true loss of mass.

Technical Corrections:

1. Many places: Apostrophes are not used correctly and there are spelling and grammar errors.

   We revised the whole paper for correct grammar and to eliminate spelling errors.

2. Abstract, line 16: Reword "look into Greenlandic discharge"

Reworded to:
In addition we examine Greenlandic discharge,...

3. Abstract, line 22: Which "data-sets"? Be specific for clarity.

   As the systematic biases come from the reanalysis and ocean-reanalysis datasets we added those.

4. Line 169: "reference salinity" should read "reference density" I think.

   That's correct, we changed it.

5. Figure 2: What are R and \mu in Figure 2 legends?

   R is runoff and \mu is the long-term mean. We added them to the figure caption.

**Response to Anonymous Referee #2**

General comments:

The manuscript is well organized, but there are some spelling and grammatical errors that needs to be considered, including the use of commas and apostrophes. I also suggest to avoid using words such as "spurious", "huge", "clearly" etc., especially for the results and conclusions sections (see also specific comments).

We revised the whole paper and correct spelling and grammatical errors and we made sure to avoid words as "spurious", "huge", "clearly",...

Considering that previous studies focusing on the Arctic drainage basin have used different approaches and motivations for its geographical domain, I am missing a motivation for the chosen boundary of the Arctic Ocean drainage basin in this study, and why e.g., Hudson Bay, and James Bay was not included? (e.g., L302-305).

*We added a clarification at line 121:*
"Figure 1 presents the study domain. As there is no strict boundary to the south, the definition of the Arctic's geographic extent varies between past studies and there is no general rule whether to include Greenland and the Hudson Bay or not. We chose our study domain to be consistent with Tsubouchi et al. (2012) as we wanted to compare the oceanic fluxes from ocean reanalysis with the observation-based estimates from the ARCGATE project. The Arctic Ocean is bounded by the position of hydro-graphic moorings in the main gateways. ..."

Clarify also in L313 that total drainage area refers to the area for this study.

We clarified this.

How do these reanalysis products take frozen components of the freshwater system into consideration, e.g., glaciers and permafrost, considering that many of the river basins in the study are underlain by permafrost? For example, lines 497-498 includes an interesting aspect that I would like to see more elaboration on.

*We added an explanation on the representation of frozen land components in reanalyses in the data section (section 2):*
"Groundwater storage is not represented in ERA5 and ERA5-Land and also the representation of frozen land components is not ideal in HTESSEL, as glaciers are depicted as large amounts of snow which are kept fixed to 10 m of snow water equivalent. When melting conditions are reached, the snow produces a water influx to the soil and consequently contributes to the total runoff. However, the mass balance is not accounted for over glaciers as the snow is restocked to constantly stay at the fixed 10 m level and hence changes in the glacial storage component cannot be assessed properly. The soil water content includes liquid as well as frozen components and thus also includes permafrost. When the soil temperature reaches melting conditions, the soil water contributes to sub-surface runoff and the soil water storage declines. However, a recent study by Cao et al. (2020) concluded that ERA5-Land soil data are not optimal for permafrost research, due to a warm bias in soil temperature that leads to an overestimation of the active-layer thickness and an underestimation of the near-surface permafrost area. Therefore, we additionally include GRACE (Gravity Recovery and Climate Experiment) satellite data, as land water storage from GRACE includes changes in soil moisture (including permafrost), glaciers, snow, surface water, aquifers and groundwater.'

*Additionally, we justified the use of ERA5 land water storage for certain purposes at lines 497-498:*
"Oceanic transports out of the Arctic domain exceed the atmospheric moisture entering the Arctic (6294 +/- 121 km3) by nearly 5%, indicating an annual loss of water volume of roughly 300 km3. The bulk part of this loss is generated through terrestrial water mass losses. Even though the representation of frozen land components is not ideal in HTESSEL, the comparison of GRACE mass changes

to the sum of ERA5 storage changes (snow and soil water) and glacial changes taken from literature (e.g., Wouters et al., 2019) agree well. Therefore, we combine land storage changes from ERA5 (excluding glaciers) with storage changes from GRACE to estimate contributions of different terrestrial sources to the diagnosed storage decline in the Arctic. We find that approximately 50% of

the 266 km3yr-1 decline are generated through liquid and solid discharge from Greenland, while about 40% come from Arctic glaciers (excluding Greenland) and the remaining 10% are the result of a decline in land water storage due to permafrost and snow cover reduction."

New references:
Cao, B., Gruber, S., Zheng, D., and Li, X.: The ERA5-Land soil temperature bias in permafrost regions, The Cryosphere, 14, 2581–2595, https://doi.org/10.5194/tc-14-2581-2020, 2020.

In the conclusions, I am missing a general discussion on implications for future studies and assessments of freshwater budgets of the Arctic Ocean.

*This question is similar to the 2. general comment by Anonymous Referee #1. We revised the whole conclusion section. Some of the key points that were added are the following:*
"Summarizing we refined past Arctic water budget estimates (e.g., Serreze et al., 2006; Dickson et al., 2007) and their uncertainties by combining some of the most recent reanalyses data-sets and observations, and by applying a variational optimization scheme. The variational adjustment worked very well on an annual scale and brought reliable estimates of the volume budget terms, requiring only moderate adjustments of less than 3% for each individual term. Adjustments are considered reliable if budget closure is achievable within the respective terms error bounds and if the terms are comparable to estimates from past studies."

"Especially when calculating Pan-Arctic runoff, caution is needed. Our results show that seasonal peaks of river discharge are underestimated in almost all of the assessed reanalyses (ERA5, ERA5-Land, GloFASE5, GloFASE5L). The biggest errors are caused by inhomogeneities in the data assimilation system (ERA5 and GloFASE5) led to a great underestimation of runoff, especially in the latter half of the time series. However also reanalyses without data assimilation (ERA5-Land and GloFASE5L) were not able to reproduce the seasonal cycle of river discharge accurately. On the other hand we find distinct improvements in the new GloFASE5new product, especially when investigating seasonal cycles and long term means it features considerable enhancements compared to its precursors."

"When extrapolating observed river discharge to the whole Pan-Arctic area we found that the common method of hydrological analogy tends to underestimate the discharge peaks. We therefore advise to use river discharge observations where available and reliable runoff/discharge estimates from reanalyses (e.g., GloFASERA5new or ERA5-Land) to extrapolate discharge to the ungauged areas."

"To further refine the budget estimates, longer time series of all budget terms would be needed. For example, one could repeat the analysis using the back extension of ERA5 which goes back to 1950. There is also a new bias-corrected ERA5 data set (WFDE5, Cucchi et al., 2020), that could be examined in terms of the Arctic water budget. Further it would help to include a precipitation observation data set, preferably one that combines available satellite-based and gauge-based data sets. Concerning estimation of biases in ocean reanalyses, one could in principle draw on information from oceanographic data for comparison. A main difficulty with oceanographic data is the generally limited temporal and spatial coverage. Nevertheless, the unique form of the Arctic Ocean (as water leaves and enters only through a handful of gateways) allows relatively easy measurements of the in- and outgoing fluxes. As an example, measurements from arrays of moored instruments (like e.g., Acoustic Doppler Current Profilers, MicroCAT – CTD Sensors and Seagliders) were taken to estimate transports through the Arctic gateways using a mass-consistent framework (Tsubouchi et al., 2012, 2018). Our results however showed that the moored instruments did not measure the velocity field accurately enough to resolve the barotropic wave signal arising from temporally varying runoff (Tsubouchi, 2019) leading to errors in the seasonality of the net volume flux. A longer measuring period with an even denser monitoring network could help with this aspect."

Specific comments:

L12: I suggest to avoid the use of "spurious" and instead explain or reference to what you are referring to.

Rephrased to:
Runoff from ERA5 and GloFAS v2.1 feature pronounced declining trends, induced by two temporal inhomogeneities in ERA5's data assimilation system, and seasonal river discharge peaks are underestimated by up to 50% compared to observations.

L37: consider removing "remarkably"

We removed it.

L41-43: Consider rephrasing for clarity and also specify the part on climatological conditions.

Rephrased to:
In addition, significant portions of the rivers discharge may bypass the gauging

stations through braided channels or as submarine groundwater. Further also climatological conditions pose a hindrance to gauge measurements, as temperatures in the northern latitudes often lead to river freeze up in late autumn and flooding in spring due to river-ice break up (Syed et al., 2007).

 L45: avoid using "huge"

We rephrased it.

 L47-48: Consider rephrasing for clarity. and
 L48-49: This is not very clear, please explain what you mean by "spurious" (see also previous comment related to this).

We rephrased it to:
However data assimilation systems can introduce biases and temporal discontinuities, as changes in the observing system are sometimes inevitable and may lead to inhomogeneities in the time series. One known change is the introduction of the IMS (Interactive Multisensor Snow and Ice Mapping System) snow product in ERA5, which led to a negative shift in ERA5's snowmelt and consequently also runoff (Hersbach et al., 2020; Zsótér et al., 2020).

 L92: Which 16 rivers were included in the study, and how was the shorter observational records treated for the analysis in comparison to the longer observational records?

We added the names of the 16 rivers:
Pur, Taz, Khatanga, Anabar, Olenek, Yana, Indigirka, Alazeya, Abadyr, Kobuk, Hayes, Tana, Tuloma, Ponoy, Onega, Mezen

And clarified the consideration of shorter vs longer records:
We calculated an observation-based Pan-Arctic river discharge for the whole period of 1979 to 2019, by calculating discharge for every time step (= every month) separately while using all river discharge measurements available at this certain timestamp. The total Pan-Arctic discharge is then obtained by calculating river discharge for the ungauged area at each individual timestamp (using two different calculation methods - see section 4.2) and adding it to the observed discharge.

 L119-120: What about frozen storage components, such as glaciers?

We added permafrost and glaciers as those were considered too through GRACE. (See answer of general comments)

 L134: I suggest to add references to earlier studies, and revise "popular" to "common" – if this is what you are referring to?

Yes, we changed the wording and added a couple references:
A common way to calculate the oceanic freshwater budget is through the assumption of a reference salinity (e.g., Serreze et al., 2006; Dickson et al., 2007; Curry et al., 2011; Haine et al., 2015).

 I suggest to remove a, b, c in subheadings (e.g., L196, 205).

We removed those.

 L244: consider removing "clearly"

We removed it.

 Fig 4: Is this figure only considering the shorter time series of the 16 catchments, or for the full time period (1981-2019)? Same question for figure 6 and the observed Pan-Arctic river discharge data.

All lines in figure 4 consider the shorter time series from 1981 to 1999.
The "observed" Pan-Arctic river discharge in fig. 6 was calculated for the period of 1979-2019, using observations where available and correction factors from GloFAS_E5new for the respective ungauged areas.
We added all time periods into the figure captions.

 L497-498: This is an interesting aspect that I would like to see more elaboration on.

See answer to general comments

 L527: I suggest to include references here, and do you mean "common" rather than "popular"?

Yes, we again changed popular to common and added references (e.g., Shiklomanov and Shiklomanov, 2003).

 L528: How does this result compare to other studies?

*We a comparison to other studies:*
"We estimate Pan-Arctic river discharge from gauge observations using monthly correction factors from GloFASERA5new, as the common method of hydrological analogy (e.g., Shiklomanov and Shiklomanov, 2003) tends to underestimate the high flow summer peaks (see Fig. 4) and obtain a long-term annual flux of 4031km3 ± 203 (excluding Greenland). To compare our results to past studies we adapted the time periods and areal extents accordingly and found reasonable accordance with Haine et al. (2015), who combined runoff from ERA-Interim with river discharge observations and obtained a total discharge about 5% higher

than our estimate. An even better agreement was found with the estimates made by Shiklomanov et al. (2021a), as the total Pan-Arctic discharges (including Greenland) agree within 2%."

L529: Runoff from ERA5 is substantially "too low" – do you mean "underestimated compared to observed discharge" or similar?

Yes, that's what we meant. We rephrased it accordingly.

L531: Please consider rephrasing and describe the "unrealistic" aspects.

*We rephrased it:*
"Those strong declines are caused by two inhomogeneities (1992 and 2004) in ERA5's snow melt time-series and contradict the discharge increases found in gauge observations. Those inhomogeneities are caused by a loss of snow through changes in the data assimilation system."

L548: What is considered "trustworthy" here – please explain.

*By trustworthy we mean the data products we have most confidence in after comparing them amongst each other and to observational datasets. We changed the sentence to the following:*
"Comparing the estimates of freshwater input into the Arctic Ocean that we have most confidence in after the preceding analysis (listed in table 7), to oceanic volume transports through the Arctic gateways computed from ocean reanalysis yields..."

L555: What is considered "reliable" – please explain

*We added an explanation:*
Adjustments are considered reliable if budget closure is achievable within the respective terms error bounds and if the terms were comparable to estimates from past studies.

L559: What would be a full success here, please elaborate.

*A full success would include elimination of the budget residuals for every single month, while at the same time staying inside the respective a priori spreads of the individual terms. However, we changed the sentence to the following:*

On a seasonal scale however, stronger adjustments were needed to close the budget, and some of the adapted fluxes fell out of their a priori uncertainty range, suggesting an underestimation of the specified uncertainties. The latter is very likely caused by the presence of systematic errors being present in the data

sets, or at least in their seasonal cycles, that are not taken into account in our a priori uncertainty estimates

 L560: revise month to months

We changed it.

 L571: Please specify what you refer to with "in most reanalyses".
 L572: Please specify what you refer to with "spurious signals".

*We changed it to:*
"Our results show that seasonal peaks of river discharge are underestimated in almost all of the assessed reanalyses (ERA5, ERA5-Land, GloFASE5, GloFASE5L). The biggest errors are caused by inhomogeneities in the data assimilation system (ERA5 and GloFASE5) which led to a great underestimation of runoff, especially in the latter half of the time series. However also reanalyses without data assimilation (ERA5-Land and GloFASE5L) were not able to reproduce the seasonal cycle of river discharge accurately."

---

## Author Response (AR2)

Dear Reviewer,

Thank you for your comments, you addressed some important points. You can find our reply enclosed.
We really appreciate your time and insight in reviewing our manuscript!

Kind regards,

Susanna (on behalf of all co-authors)

General comments

There is often a comma missing when the sentence starts with "However", "Hence", "Further", "Furthermore" etc. (see e.g., L36, 75, 149, 161, 163, 180, 200, 239, 251, 259, 309 etc., and other places in the text) Please check this throughout the manuscript.

Thanks, we revised the whole text and added commas where necessary.

Please make it clear in the figure captions what period the time series are referring to, especially in those cases where graphs show different time series, e.g., Figure 6.

We adapted the figure captions accordingly and changed the labeling of the graphs to make them clearer.

If you by fluxes refer to freshwater fluxes, please indicate this in the text.

We specified whether the mentioned fluxes are freshwater of volume fluxes.

Specific comments

L21: Consider revising to "However, on a seasonal scale budget residuals…."

L24: Change "that" to "which" or remove the comma.

L40: Revise to Rivers'

L41: remove "also", e.g. "Further, climatological conditions pose a hindrance…."

L88: Add comma "In contrast, discharge…"

L91: Add comma "For the Pan-Arctic approach, river discharge…"

L136: Add comma "In addition, volumetric fluxes…"

L137: Revise to "observation-based estimates"

L161: Move e.g., to the beginning (e.g., Serreze et al….)

L165: Add comma after "budgets"

L205: Is "Hence" needed here?

L228: Consider removing "in order" at the beginning of the sentence and start with "To" – and – do you mean "we follow methods by Mayer…"

L267: Change heading to "4 Results and Discussion"

L275: Add comma before "as well as"

L276: Add comma before "and weak runoff…"

L301: Revise to "but is mostly caused due…"

L302-303: Consider revising: "For example, at the upper part of the Ob river, ice breaks up around April to May, while the lower part breaks up between May and June (Yang et al., 2004b)."

L310: Add comma "In autumn and winter, land water…"

L545: Add comma after summer "In late summer, river discharge…"

L596: suggest to revise "too low" to "underestimated" – and revise "errors" to "discrepancies"

L608: Suggest to remove "(see next bullet point)"

L629: Suggest to revise to "In summary, …"

L636, 647, 654: Consider revising "quite"

L838: Muntjewerf et al. 2020 should come after Muñoz Sabater et al. 2019

We revised, corrected, or rephrased all the points above.

L96-98: This sentence is not clear to me. By "separately" – do you mean the monthly time step or for each river? With "certain timestamp" – are you referring to the 1979-1999 period?

By separately we meant the monthly time step and with "at this certain timestamp" we referred to each of those time steps. We reformulated the sentence to the following:

*We calculated an observation-based Pan-Arctic river discharge for the whole period of 1979 to 2019, by calculating discharge separately for every time step (= every month), using all river discharge measurements available at those time steps.*

Figure 1: Is it possible to also include the additional 20 river basins, or are they too small to display on the map?

Yes, we included the additional basins.

Figure 3: The graph appears to be cut-off at the bottom. Please make it clear in the figure or caption what the x-axis (80-20) represents (e.g., 1979-2019).

We added the time period in the caption and also adapted the x-axis to full years (1980 etc.).

Table 4-5: Why is there a difference in time period compared to Figure3 (1979-2019)?

That's because ERA5-Land data are not available before 1981, hence all trends are calculated over the coinciding period 1981-2019 (we mention this in line 260 of the manuscript). Figure 3 shows the fully available timeseries of all datasets and hence most of the lines go back to 1979.

L378: Could maps be shown in appendix?

Yes, we added maps for GRACE land and ocean water storage changes to the appendix.

We further found some strong land to ocean leakages in the GRACE data that we used, and therefore we exchanged the GRACE data using spherical harmonics solutions to the GRACE Mascon solutions, where the leakage effect is considerably reduced. This led to changes in some figures, tables, the variational optimization, and the corresponding text passages. However, we note that the main results did not change considerably, and the outcome of our study was not affected. The biggest difference was a change of the sign of the oceanic water storage, as we now get an increase in water

mass stored in the Arctic Ocean. We therefore adapted the discussion of those results to the following (Line 552 in the revised manuscript):

*The reported increase in oceanic storage is driven by mass increases over the western Arctic Ocean and the coastal areas of Eurasia and North America (see Fig. A1 in the appendix). For the western Arctic Ocean various studies (e.g., Proshutinsky et al., 2019) indicate an accumulation of freshwater in the Beaufort Gyre, due to a combination of favourable wind forcing, redirection of Mackenzie River discharge, inflow of low salinity waters through Bering Strait and sea ice melt. Mass increases along the coastal areas are the result of runoff increases. Mass decreases in Barents and Kara Sea , as well as in Baffin Bay (Fig. A1 in the appendix) counteract those increases, resulting in the reported minor trends for the whole Arctic Ocean. Mass decreases to the west of Greenland are mainly caused due to lowering of the geoid associated with nearby ice mass losses, however Jeon et al. (2021) also found residual land-leakage effects that were not removed in the GRACE Mascons. Furthermore, oceanic storage is strongly affected by decadal wind variations (Volkov and Landerer, 2013; Fukumori et al., 2015) and circulation patterns and exhibits strong nonseasonal fluctuations, further aggravating the detection of real oceanic mass trends. Volkov and Landerer (2013) found that an intensification of the westerly winds over the North Atlantic and over the Russian Arctic continental shelf lead to a decrease of ocean mass in the central Arctic. Further, they found positive correlation between Arctic Ocean mass fluctuations and northward wind anomalies over the Bering Seas and the northeastern North Atlantic. They also revealed that cyclonic/anticyclonic anomalies of the large-scale ocean circulation lead to negative/positive Arctic ocean mass anomalies.*

Figure 6: Make it clear in the figure caption what time period right graphs are referring to.

We added the time periods in the caption.

Figure 12: Is this figure representing the fluxes of the hydrological cycle? If so, consider revising to – 1993-2018 adjusted long term means of freshwater fluxes in the Arctic hydrological cycle. Units are km3 per year; arrows are scaled by the magnitude of the freshwater flux.

Yes, however the oceanic fluxes represent volume instead of freshwater fluxes. We changed the caption to the following:

*1993-2018 adjusted long term means of atmospheric and terrestrial freshwater fluxes and storage terms in the Arctic hydrological cycle, as well as oceanic volume transport and storage change. Units are km3 per year; arrow areas are scaled by the magnitude of the represented terms.*

Figure 13: Caption – by cycle – do you mean freshwater fluxes? (see comment on Figure 12), e.g., "1993-2018 adjusted mean annual freshwater fluxes of the Arctic Ocean's hydrological cycle. Shading represents the uncertainties of the a priori estimates." Consider to update the text in related sections accordingly.

We changed the caption to the following:

*1993-2018 adjusted mean annual cycles of atmospheric and terrestrial freshwater fluxes in the Arctic hydrological cycle as well as oceanic volume transports and storage changes before (left) and after (right) the optimization. Shading represents the uncertainties of the a priori estimates.*

We also adapted the figure to include the seasonal cycles before and after the optimization and added lines showing the seasonal residual, this did not change any results and just made the figure easier to understand for the reader.